# Coordination of alternative splicing and alternative polyadenylation revealed by targeted long read sequencing

Zhiping Zhang[1,2], Bongmin Bae [2], Winston H. Dredge [2] & Pedro Miura [1,2,3] ✉

Nervous system development is associated with extensive regulation of alternative splicing (AS) and alternative polyadenylation (APA). AS and APA have been extensively studied in isolation, but little is known about how these processes are coordinated. Here, the coordination of cassette exon (CE) splicing and APA in *Drosophila* was investigated using a targeted long-read sequencing approach we call Pull-a-Long-Seq (PL-Seq). This cost-effective method uses cDNA pulldown and Nanopore sequencing combined with an analysis pipeline to quantify inclusion of alternative exons in connection with alternative 3' ends. Using PL-Seq, we identified genes that exhibit significant differences in CE splicing depending on connectivity to short versus long 3'UTRs. Genomic long 3'UTR deletion was found to alter upstream CE splicing in short 3'UTR isoforms and ELAV loss differentially affected CE splicing depending on connectivity to alternative 3'UTRs. This work highlights the importance of considering connectivity to alternative 3'UTRs when monitoring AS events.

mRNA transcripts are subject to a variety of co/post-transcriptional processing events in metazoan cells including alternative splicing (AS) and alternative polyadenylation (APA). These events are highly regulated during development and cell differentiation, including during neuronal differentiation[1,2]. More than 70% of protein-coding genes in mammals and about half in flies harbor more than one functional polyadenylation site (polyA site)[3–5]. APA occurring in the terminal exon that alters the length of the 3' untranslated region (3'UTR) is called tandem 3'UTR APA[6]. Transcripts of APA-regulated genes cleaved at the distal polyA sites are highly enriched in the nervous system[5,7,8]. In *Drosophila*, the neuron-specific RNA binding protein Embryonic Lethal Abnormal Visual System (ELAV) is the major determinant of long 3'UTR expression in neurons via APA regulation[9–13]. In mammals, roles in neural-specific 3'UTR lengthening have been proposed for the ELAV-related Hu proteins and PCF11[14,15]. Neural enriched long or extended 3'UTR mRNA isoforms have greater potential for regulation by RNA binding proteins (RBPs) and microRNAs. Long 3'UTR isoform-specific functions in the nervous system include dendrite pruning, axon outgrowth,

reproductive behavior, and neural plasticity[16–21]. Disruption of APA regulators is involved in human disease, in particular members of the Cleavage Factor I and II complexes[14,22–24]. Mutations that cause the loss or gain of poly(A) sites also underlie several human disorders[25]. Genetic disruptions in 3'UTRs that affect APA are risk factors for brain disorders, such as *SNCA* in Parkinson's disease[25,26].

ELAV widely regulates alternative splicing in the nervous system in addition to its role in regulating APA[10,13]. In fact, multiple RBPs that regulate AS have also been found to regulate APA[27–30]. In addition, Cleavage and Polyadenylation factors, which can regulate APA, have been reported to bind coding regions to facilitate splicing[31]. Additional support for shared mechanisms of regulation comes from correlations between APA and AS identified on a transcriptome-wide level[32,33]. Understanding how AS and APA events are connected within mRNAs requires a sensitive sequencing method that can provide an abundance of reads long enough to span from upstream of an alternative exon to the end of long 3'UTRs.

In contrast to widely employed short-read high-throughput sequencing technologies that generate reads <150 nt in length, long-

[1]Department of Genetics and Genome Sciences, University of Connecticut School of Medicine, Farmington, CT, USA. [2]Department of Biology, University of Nevada, Reno, Reno, NV, USA. [3]Institute for System Genomics, University of Connecticut, Storrs, CT, USA. ✉e-mail: miura@uchc.edu

read sequencing techniques enable sequencing of full-length transcripts up to dozens of kilobases (kb)[34]. Long read RNA-Seq is most commonly performed on Pacific Biosciences (PacBio) single-molecule real-time (SMRT) and Oxford Nanopore Technologies (ONT) nanopore platforms[35–39]. Long-read RNA-Seq holds great potential for understanding how different types of co/post-transcriptional processes are orchestrated across developmental stages, tissues, and cell types. Nascent RNAs have been examined using long-read sequencing technology to investigate how various RNA processing events are coupled[40] and there is evidence for global coordination between the efficiency of co-transcriptional splicing and 3′ end processing within individual transcripts[36].

A major drawback to transcriptome-wide long read RNA-sequencing is that full-length read information is mostly restricted to high abundance and relatively short transcripts. Enrichment strategies prior to long-read sequencing can enable sufficient depth of coverage for a targeted subset of genes. Several groups have successfully performed large-scale probe-based cDNA capture followed by PacBio long-read sequencing[41–43]. Others have employed cDNA capture coupled to long-read sequencing for smaller sets of genes on the nanopore platform[44,45]. These studies have generally focused on new isoform discovery and exon connectivity, in particular AS. Fewer studies have employed these long-read approaches to quantify how alternative exons connect to alternative 3′UTRs regulated by APA[33,46]. This is a technically challenging problem given the especially long length of many 3′UTRs and the need to distinguish between tandem APA 3′UTRs that share a common region.

Our previous work showed that for the *Dscam1* gene in *Drosophila*, AS of an upstream cassette exon (CE) was strongly influenced by whether it was connected to a long or short 3′UTR[18]. ELAV is a regulator of both AS and APA for many genes, including *Dscam1*[9–13,18]. Long 3′UTR deletion and minigene reporter analysis showed that regulation of *Dscam1* CE splicing by ELAV required the presence of the long 3′UTR mRNA isoform[18]. Here, we set out to identify coordination of CE splicing and alternative 3′ end processing during *Drosophila* embryonic development. To accomplish this, we developed a targeted Nanopore long-read sequencing approach we call Pull-a-Long-Seq (PL-Seq). This approach facilitated the study of AS and APA coordination in a cost-effective and efficient manner. We used PL-Seq to identify 23 genes that exhibit 3′UTR connected AS in neuron-enriched tissues and quantify how ELAV regulates coordinated AS-APA. We also examined the cross-talk between AS and APA by genomic alteration of these events in *Drosophila* and quantifying their impact on each other.

## Results

### 3′UTR lengthening is significantly associated with CE regulation during embryonic development

Our previous work uncovered that coordinated AS and APA occurs during embryonic development for the *Dscam1* gene[18]. Browsing embryonic development short read RNA-Seq tracks[47] we identified another gene, *Khc-73*, that also shows coordinated upstream CE alternative splicing and 3′UTR lengthening (Fig. 1a). In this case, later developmental stages show increased inclusion of two upstream CEs (exons 12 and 15) that coincided with expression of the long 3′UTR isoform. We set out to identify more genes that undergo coordinated AS and APA in *Drosophila*. QAPA, a tool that enables estimation of alternative polyA site usage (PAU) of tandem APA events[48], was used to quantify distal polyA site usage (dPAU) throughout embryonic development and in several dissected tissues. The distribution of dPAU across developmental stages and tissues revealed many genes shifting to distal polyA site usage later in development and in the nervous system, which is consistent with previous published observations (Fig. 1b)[5]. We compared the usage of distal 3′UTRs before (2–4 h embryos) and after (16–18 h embryos) establishment of the nervous system. Among 1951 expressed genes with multiple APA isoforms, 252

genes exhibited greater expression of the most distal 3′UTR in the later stage (lengthening), whereas 42 showed the opposite trend (shortening) (Fold Change > 2 & $p < 0.05$) (Fig. 1c)(Supplementary Data 1).

AS in the *Drosophila* central nervous system is known to be widespread[49]. Using rMATS[50], we detected a variety of regulated AS events between 16–18 h and 2–4 h embryos. CE splicing was the largest group of AS events, with 358 exon skipping and 458 exon inclusion events affecting a total 446 genes ($|\Delta\text{PSI}| > 0.2$ and FDR < 0.05) (Fig. 1d). For the 252 genes exhibiting 3′UTR lengthening during embryonic development, 58 also harbored one or more differentially regulated CEs (Fig. 1e). These 58 genes showed no preference for increased inclusion or skipping in the later developmental stage (58 CEs increased skipping, 82 CEs increased inclusion, $p = 0.6448$, two-sided Fisher's exact test). When compared with all the genes subject to APA in embryos, Fisher's exact test showed that these 3′UTR lengthening genes were significantly associated with concurrently regulated CE splicing events (AS-APA) in the same host gene (Fig. 1e, $p = 1.519\text{E-}07$). The CE splicing and APA developmental regulation trends for two of these genes, *Khc-73* and *Dys*, was confirmed using RT-PCR (Supplementary Fig. 1). Gene Ontology analysis revealed multiple categories of enrichment for the 58 genes, including molecular functions of phosphatase activity and receptor binding and biological processes of axon guidance (Fig. 1f).

For many APA-regulated genes, isoforms using the distal polyA site are not highly expressed until adulthood in the brain (Fig. 1b). Thus, we performed an additional pair of comparisons between adult head and ovary samples, since the ovary has been shown to generally lack long 3′UTR expression[5]. When compared with ovaries, 290 genes were found to have increased expression of the long 3′UTR isoform in heads, whereas only 8 showed the opposite trend (Supplementary Fig. 2a). Among these 290 3′UTR lengthening genes, 62 were found to also harbor one or more differentially regulated CE in heads compared to ovaries (Supplementary Fig. 2b, Supplementary Data 2). A significant association between 3′UTR lengthening and differentially spliced CEs was also revealed by Fisher's exact test (Supplementary Fig. 2c, $p = 0.0002$).

### PL-Seq reveals pairing of polyA site choice with CE alternative splicing for 23 genes

We wanted to understand how AS changes detected for the APA-regulated genes are distributed between different 3′UTR isoforms. One might expect that if a given gene undergoes both AS and APA during embryonic development, the regulated CE could be 3′UTR independent or show biased incorporation depending on the 3′UTR isoform. We performed an RT-PCR-based nanopore sequencing approach targeted specifically for the *Khc-73* gene, which we previously developed for *Dscam1*[18]. We generated RT-PCR amplicons representing all 3′UTR isoforms that also capture the alternative splicing events for exons 12 and 15 (Uni), as well as amplicons that represent those associated with the extended 3′UTR sequence (Long). Nanopore sequencing revealed a significant difference in the PSI for both exons 12 and 15, with greater exon inclusion observed in the Long vs Uni amplicons (Supplementary Fig. 3a, b). This suggests that AS of these exons is connected to 3′UTR choice. Caveats of this approach include that it does not reveal the CE alternative splicing pattern specifically in short 3′UTR isoforms and the PCR amplification is performed for only a single gene.

Current long-read sequencing approaches performed transcriptome-wide are inadequate for obtaining sufficient depth of long reads for the targets of interest to accurately quantify the connectivity of AS-APA events. To address this, we developed a probe-based cDNA pulldown strategy to enrich for genes of interest prior to sequencing on the Nanopore platform (Fig. 2a). We call this method Pull-a-Long Seq (PL-Seq). In this approach, SMARTer cDNA synthesis using oligo (dT) priming is performed to obtain full-length cDNA from total RNA, and SMARTer oligo sequence is introduced at 5′ and 3′ ends

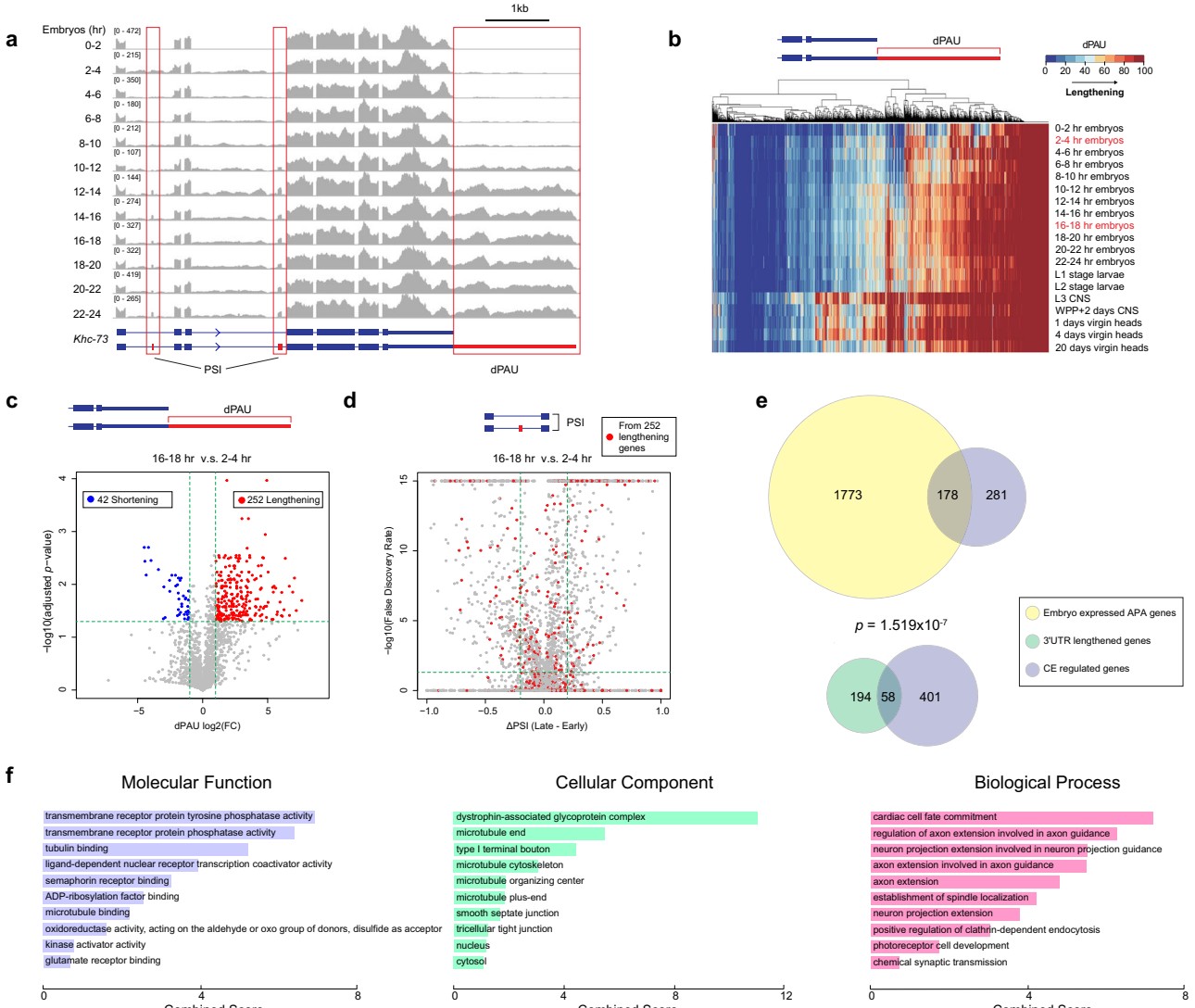

**Fig. 1 | APA and AS analysis of short-read RNA-Seq data across development and tissues in *Drosophila*. a** Short-read RNA-Seq data[47] shows coordinated inclusion of CEs with 3'UTR lengthening during embryonic development for the *Khc-73* gene. **b** Distribution of Distal PolyA Site Usage (dPAU) values from different embryonic stages and larval/adult tissues. The value of dPAU falls into a range from 0 (no distal polyA site usage) to 100 (exclusive distal polyA site usage). **c** dPAU compared between late-stage embryos (16–18 hr) and early-stage embryos (2–4 hr). *p* values are calculated from two-tailed t-test, and then adjusted using FDR. Blue represents FDR adjusted *p* < 0.05 and FC < 0.5 while red represents adjusted *p* < 0.05 and FC > 2. FC, fold change. **d** Change in PSI compared between late-stage embryos (16–18 h) and early-stage embryos (2–4 hr) for CEs. PSI ranges from 0 to 1, and ΔPSI is calculated as PSI (16–18 h) – PSI (2–4 h). Red dots represent CE events from the 252 3'UTR lengthening genes identified in (**c**). Horizontal dash lines in (c) indicate adjusted *p* = 0.05 and in (d) indicate FDR = 0.05. Vertical dash lines in (**c**) indicate FC = 0.5 (left) and FC = 2 (right) and in (**d**) indicate ΔPSI = −0.2 (left) and ΔPSI = 0.2 (right). **e** Fisher's exact test (two-sided) showing 3'UTR lengthening genes are significantly associated with regulated CE events in late vs early-stage embryos. **f** Gene Ontology analysis of the 58 genes exhibiting 3'UTR lengthening and regulated CE during embryonic development. Also see supplementary Figs. 1, 2.

to enable downstream PCR amplification. Two to five biotinylated probes are designed to enrich each target gene, with the probes targeted to constitutive exons and universal 3'UTR sequences (Supplementary Data 3). After PCR amplification and library preparation, Nanopore sequencing is performed on the MinION sequencer.

A series of alignment and gene-specific filtering steps are required to quantify the upstream exon inclusion for long versus short 3'UTRs. First, alignment to the *Drosophila* genome (dm6) is performed using minimap2[51], and then reads from a given experiment are filtered for regions spanning alternative CEs and alternative 3' ends for the targeted genes. We outline these filtering steps for the *Khc-73* gene as an example (Fig. 2b). Reads are selected that cover the common 3'UTR region and exons that flank the CE of interest– in this case, the reads are required to include the constitutive exons 11 and 13 (Fig. 2b). Reads which span the extended 3'UTR region are selected. These reads serve

exclusively as "Long 3'UTR" parsed reads. Then, the remaining reads are filtered to account for mispriming from genomically encoded A stretches, and reads containing untemplated polyA tails are selected to constitute the "Short 3'UTR" parsed reads. Percent-Spliced-In (PSI) values of upstream CEs can then be exclusively assigned to each 3'UTR mRNA isoform. Individual full-length reads from late-stage embryos demonstrate a preferential inclusion of exons 12 and 15 in the long 3'UTR reads compared to the short 3'UTR reads (Fig. 2c, d). This difference is even more striking when observing filtered coverage tracks (Fig. 2d). For both 16–18 hr embryos and adult heads there is a near binary switch in the usage of exons 12 and 15 in long versus short 3'UTR isoforms (Fig. 2c, d). The *Khc-73* long 3'UTR isoform exhibited an exon 15 PSI of 96.8% compared to 1.6% for the short 3'UTR isoform (Fig. 2c). Similar results were obtained from adult heads (Fig. 2d).

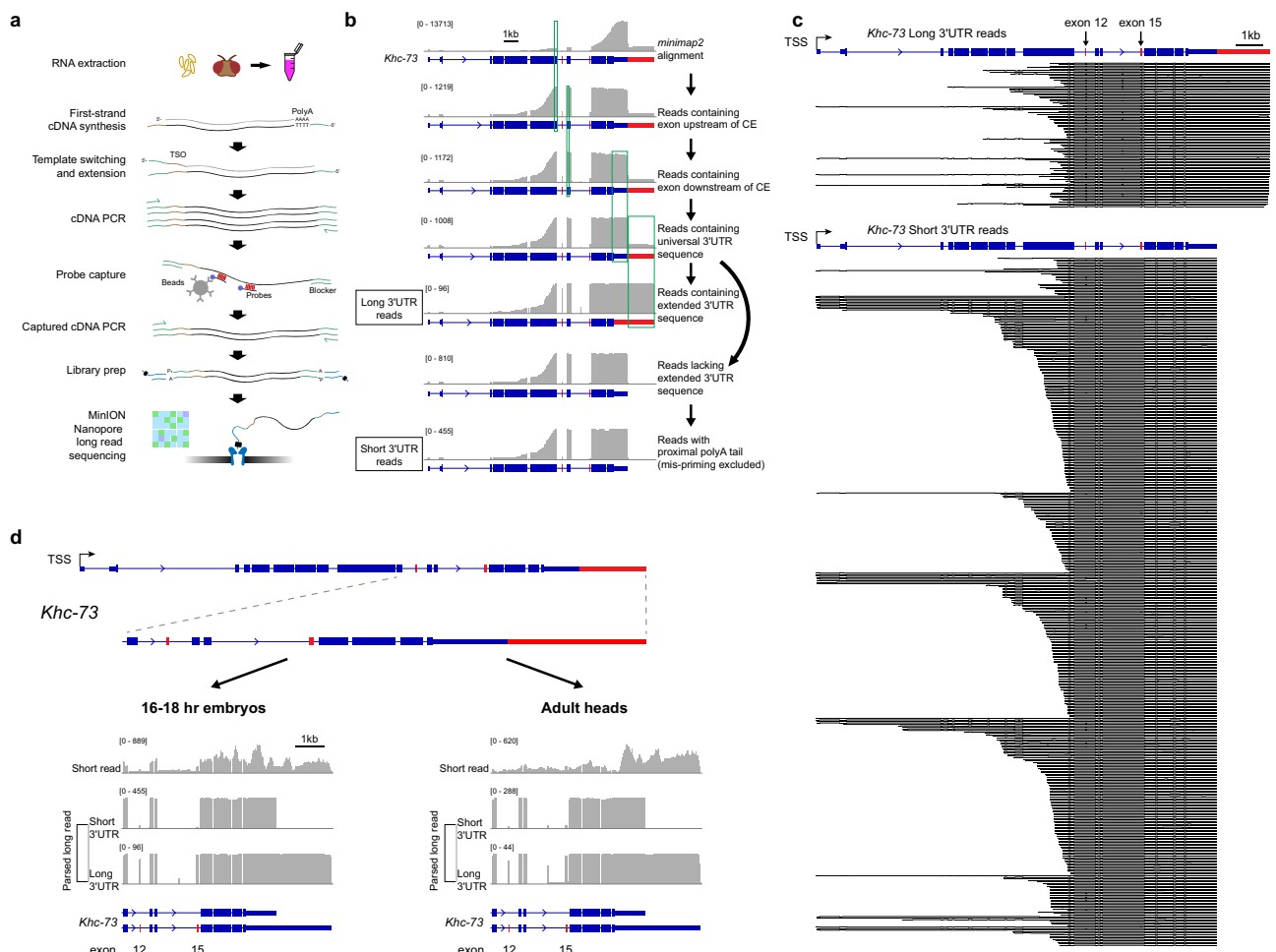

**Fig. 2 | PL-Seq library preparation and data processing workflow. a** Illustration showing PL-Seq library preparation. **b** Filtering of reads and parsing to long and short 3'UTR isoforms demonstrated on the *Khc-73* gene. Reads aligned to *Khc-73* are filtered to ensure that they contain upstream and downstream constitutive exons of each specific CE event, and the universal 3'UTR sequence and then parsed to long or short 3'UTR reads. **c** Parsed long reads from *Khc-73* long (top) and short (bottom) 3'UTR isoforms in a 16–18 hr embryo sample. **d** Coverage tracks from short-read RNA-Seq (top track) and PL-Seq (bottom tracks) showing contrasting CE splicing patterns between parsed short and long 3'UTR reads for *Khc-73* in late-stage embryos (left) and adult heads (right). TSS, Transcription Start Site.

To examine the efficiency of the probe-based cDNA capture, we compared embryo PL-Seq libraries prepared with pulldown for 15 genes vs without pulldown. Reads from the 15 targeted genes comprised 0.05% of reads without pulldown. After pulldown, these 15 genes accounted for 46.43% of aligned reads, demonstrating the effectiveness of the enrichment strategy (Fig. 3a). A potential concern with the probe-based pulldown approach is that it could introduce experimental biases that alter PSI value calculation. To test for this possibility, we examined the *stai* gene, which could be sequenced at sufficient depth in the absence of enrichment by cDNA pulldown. *Stai* exon 6 PSI was found to not be altered in pulldown versus no pulldown libraries (Fig. 3b). Next, we compared read coverage across gene bodies for the no pulldown versus pulldown library. When comparing the 15 genes of interest, both libraries showed a similar bias for the 3' end, since RT is initiated with oligo dT (Fig. 3c). This bias was less evident when all genes detected ($n = 9487$) were plotted for the no pulldown condition, most likely due to the 15 genes of interest generating particularly long mRNAs. A larger portion of novel splicing junctions were observed in the pulldown sample versus the control library (Supplementary Fig. 4). These was expected given the relatively long length and complexity of the genes targeted for pulldown. Read length distribution was found to be skewed longer in the pulldown library, also reflecting the pulldown of longer cDNAs from the target genes (Fig. 3d).

We generated two pulldown probe sets that targeted 31 of the 93 genes identified as having regulated AS and APA in late versus early-stage embryos and/or in adult heads versus ovaries (see Supplementary Data 3 for probe density per gene and sequences). PL-Seq data from three to five biological replicates is shown for individual CE PSI values belonging to long or short 3'UTR for 16–18 hr embryos (Fig. 3e) and adult heads (Fig. 3f). Three genes lacked sufficient read depth to quantify changes in 3'UTR specific alternative splicing. Of the remaining 28 genes, we found that 23 genes showed significantly different CE splicing between short and long 3'UTR isoforms either in 16–18 hr embryos (20 genes), adult heads (15 genes), or both (12 genes) (two-tailed paired t-test, $p < 0.05$). Only 5 of the tested genes with appropriate read coverage showed no significant 3'UTR discrepancy of CE splicing either in 16–18 hr embryos or in adult heads (Fig. 3e, f). For the 23 genes showing connected CE alternative splicing and APA, the CE PSI difference ($PSI_{long}$-$PSI_{short}$) varied widely, from −0.788 to 0.955 (Supplementary Fig. 5). Multiple genes were found to exhibit 3'UTR connected CE splicing even when they were not originally identified as AS-APA genes from short-read data. These included *pod1*, *Crag*, *Eip63E*, *shi*, and *Calx* in embryos, and *Dys* and *Dscam1* in heads (Supplementary Fig. 5). This suggests that 3'UTR connected CE splicing likely affects far more genes than those revealed to have regulated AS and APA in short read RNA-Seq data (Fig. 1e, Supplementary Fig. 1).

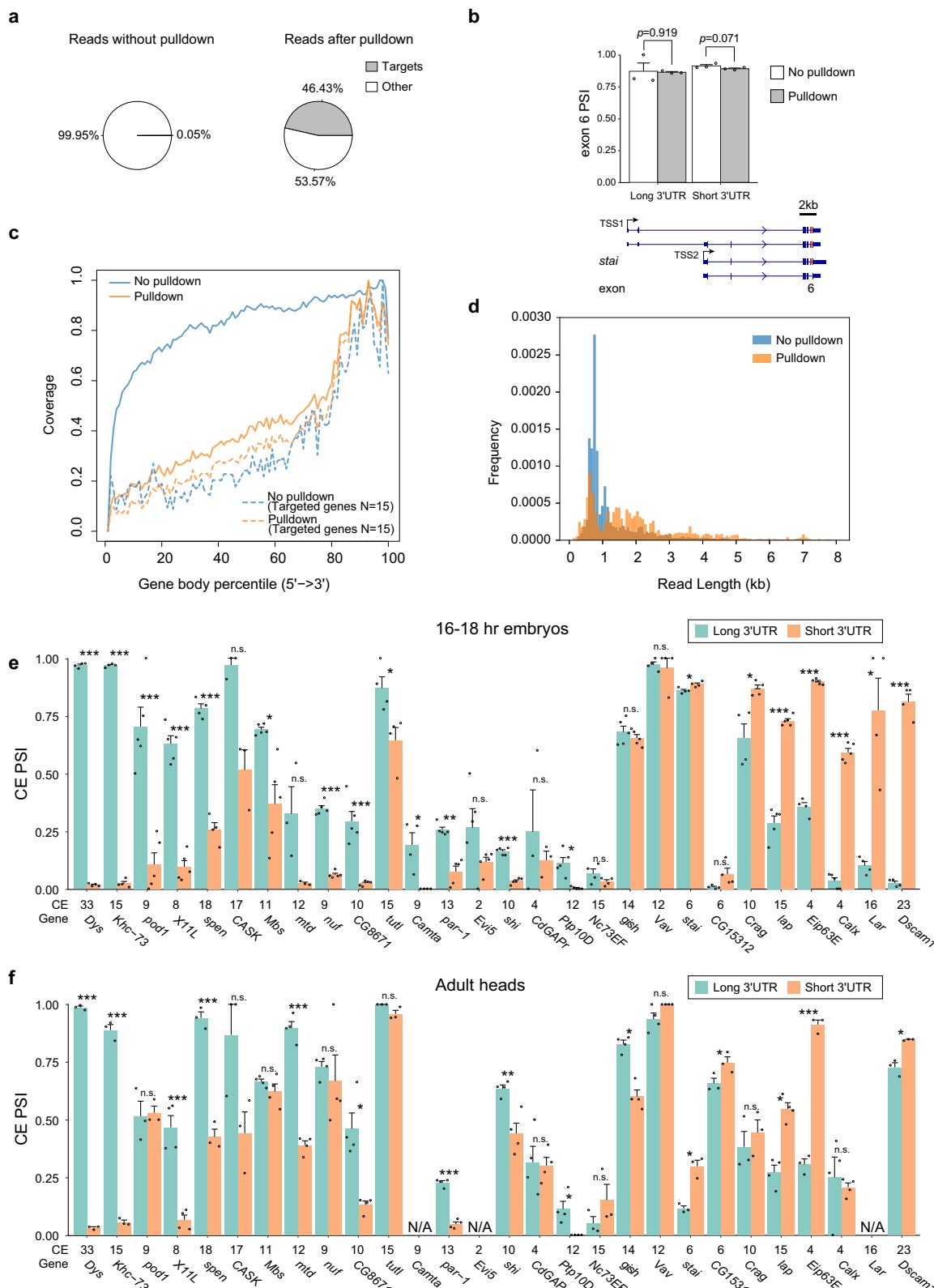

PL-Seq revealed interesting cases of connectivity between alternative exons and 3'UTRs. For some genes, trends were different in late-stage embryos compared to adult heads. We previously demonstrated the connectivity of exon skipping events to the long 3'UTR isoform of the *Dscam1* gene using a variety of methods including nanopore sequencing of "long" and "uni" PCR products (as performed for *Khc-73*, see Supplementary Fig. 3a, b)[18]. In late-stage embryos, the *Dscam1* long

3'UTR exhibits 3.7% PSI for exon 23, whereas the short 3'UTR exhibits 79.9% PSI (Fig. 4a). The PSI difference of exon 23 between long and short 3'UTR is reduced in adult heads, but remains significant. In addition, two microexons flanking exon 23 becomes highly expressed and are found to be mainly connected to the long 3'UTR in heads but not embryos. There were several additional genes, including *Khc-73* and *Dys*, that had multiple adjacent CEs included preferentially in their

**Fig. 3 | PL-Seq reveals 3′UTR-dependent CE splicing for 23 genes. a** Distribution of reads in PL-Seq libraries performed without (left) or with (right) pulldown for 15 target genes. **b** *stai* exon 6 PSI compared between "No Pulldown" and "Pulldown" libraries. For both long and short 3′UTR isoforms, exon 6 PSI shows no significant difference. $n = 3$ biologically independent samples. **c** The gene body coverage and **d** length distribution of aligned reads in No Pulldown and Pulldown samples. For gene body coverage, all transcripts are scaled to 100 and the number of reads covering each relative position is calculated. **e** CE PSIs for short and long 3′UTR

isoforms in late-stage embryos (16–18 h) from 28 genes. $n = 3$–5 biologically independent samples. **f** CE PSIs for short and long 3′UTR isoforms in adult heads. The cassette exon number for each gene is listed on the x axis (CE). $n = 3$–4 biologically independent samples. Data is presented as Mean + SEM. * indicates $p < 0.05$, ** $p < 0.01$ and *** $p < 0.005$. Two-tailed paired t- test, no adjustment for multiple comparisons. Genomic locations of CEs are found in Supplementary Data 9. Also see Supplementary Figs. 4, 5. Source data are provided as a source data file.

long 3′UTRs (Figs. 2c, d, and 4b). We previously found that *Dscam1* exon 19 is completely skipped in long 3′UTR isoforms from adult heads, but could not specifically measure this skipping in the short 3′UTR isoform[18]. PL-Seq enabled the detection of short 3′UTR-specific exon 19 PSI (98.1% in heads) (Supplementary Fig. 6a). As found previously, 0% exon 19 PSI was observed for the long 3′UTR in heads (Supplementary Fig. 6a).

For some genes with relatively shorter full-length sequences (<4 kb), we were able to obtain an abundance of reads that span the entirety of the gene from 5′ to 3′ end. For example, the *Eip63E* long 3′UTR isoform shows preferential usage of a downstream alternative first exon compared to the short 3′UTR isoform (Fig. 4c). Similarly, the use of an alternative upstream first exon for the *stai* gene occurs for the short 3′UTR isoform but is nearly non-existent in the long 3′UTR isoform (Fig. 4d). PL-Seq analysis revealed other types of 3′UTR connected alternative splicing events beyond CEs. For example, *Calx* exhibits 3′UTR connected mutually exclusive exons (MXE), and these MXEs also behave as alternatively spliced CEs (Supplementary Fig. 6b). We also observed a case for a slight shift in an upstream 3′ splice site depending on 3′UTR choice for *X11L* (Supplementary Fig. 6c), similar to what was found previously for *Dscam1*[18,52]. Together, these examples demonstrate that PL-Seq reveals the complexity of 3′UTR–linked alternative exon usage.

## Genomic deletion of long 3′UTR alters CE splicing in short 3′UTR isoforms

In previous work, we found that genomic deletion of the *Dscam1* long 3′UTR (*Dscam1*^ΔL^) altered CE splicing of exon 19 for the remaining mRNAs as measured by RT-PCR[18]. PL-Seq of *Dscam1*^ΔL^ fly heads was performed to precisely determine the CE splicing pattern of exons 19 and 23 in short 3′UTR isoforms remaining after long 3′UTR deletion. PL-Seq revealed that *Dscam1*^ΔL^ fly heads showed no expression of *Dscam1* long 3′UTR transcripts, as expected (Fig. 5a). In the remaining short 3′UTR mRNA isoforms, there was a significant reduction in exon 19 PSI (Control PSI = 92.1%, *Dscam1*^ΔL^ PSI = 66.2%, $p = 0.004$). The remaining short 3′UTR in *Dscam1*^ΔL^ fly heads thus exhibits increased skipping of exon 19. This suggests a feedback system that ensures exon 19 skipped mRNAs are expressed.

We performed a similar CRISPR/Cas9 mediated genomic deletion of the *Khc-73* long 3′UTR (*Khc-73*^ΔL^) to determine whether long 3′UTR loss could impact the exon content of the remaining *Khc-73* short 3′UTR mRNAs. Removal of the genomic region downstream of the proximal polyA site and past the distal polyA site resulted in complete loss of long 3′UTR mRNAs with flies being homozygous viable, as is the case for *Khc-73* null mutants[53] (Fig. 5b). *Khc-73* short 3′UTR mRNA isoforms normally exhibit very low inclusion of exons 12 and 15, whereas levels of inclusion are much higher in the long 3′UTR mRNAs (Figs. 2d and 5b). In *Khc-73*^ΔL^ fly heads the short 3′UTR mRNAs displayed massively increased exon inclusion for exons 12 and 15. Exon 15 PSI in the short 3′UTR mRNA was increased almost 10-fold from 5.4% to 52.7% ($p = 6.3E-07$). These PSI values approached what was found for exon 15 PSI in the WT long 3′UTR samples. Thus, both *Dscam1* and *Khc-73* short 3′UTRs exhibit alteration in CE AS upon genomic loss of the long 3′UTR region. These results suggest that alternative splicing of these exons might be influenced by 3′ end processing or the sequence content of long 3′UTRs.

Most splicing events are considered to occur co-transcriptionally[54]; thus, an alternative hypothesis is that the strong connectivity of CEs with alternative 3′UTRs is dictated by the CE event regulating downstream APA. To test this, we forced the removal of CEs in *Dscam1* exon 19 and *Khc-73* exon 15 by deleting these exons and their flanking introns. *Dscam1* exon 19 deletion was found to be homozygous lethal; thus, we performed analysis on heterozygous mutant flies. The *Khc-73* exon 15 deletion flies were homozygous viable. We measured the impact of these deletions on the relative expression of the long to short 3′UTR by RT-qPCR and for both we found this to be unchanged (Fig. 4c, d). This suggests that CE alternative splicing does not impact 3′UTR choice for *Dscam1* and *Khc-73*.

## 3′UTR connected CE alternative splicing is deregulated in *elav*^5^ mutants

ELAV and the related protein FNE are key regulators of AS and APA in the nervous system[10,13,18]. We re-analyzed short-read RNA-Seq data from L1 CNS samples of *elav,fne* double mutants versus control flies to obtain AS and APA changes[9] (Supplementary Fig. 7a, b, Supplementary Data 4). We identified 29 genes that are regulated by CE alternative splicing and 3′UTR shortening in the mutant condition compared to wild type. The 3′UTR shortening in these mutants was significantly associated with upstream CE alternative splicing regulation (Fisher's exact test, $p = 0.0004$, Fig. 6a). Out of the 29 ELAV/FNE regulated AS-APA genes, 3′UTR connected PSI data was available for 10. Eight out of 10 genes showed 3′UTR dependent CE alternative splicing in late-stage embryos and/or adult heads (Fig. 6b). In previous work, 23 genes were found to have both regulated AS and APA in *elav,fne* mutants embryos[13]. Our PL-Seq data included 7 of these genes, with 6 exhibiting 3′UTR dependent CE splicing (Supplementary Data 5). We reasoned that PL-Seq could be used to determine if CE splicing regulated by ELAV/FNE somehow depends on which 3′UTR isoform is selected downstream. *elav,fne* double mutants would have long 3′UTR isoform expression near zero for many genes, making it difficult to quantify CE alternative splicing in long 3′UTR isoforms. In contrast, from L1 CNS samples of *elav*^5^ mutants display much fewer APA changes[9] (Supplementary Fig. 7c, d). Thus, we performed experiments with *elav*^5^ mutant embryos instead, as they retain some expression of long 3′UTRs (Supplementary Data 6).

To investigate the role of ELAV in APA and CE AS connectivity, we performed PL-Seq for the previously validated 23 AS-APA genes and several control genes to monitor the *elav* mutation (Supplementary Data 3). PL-Seq data collected from *elav* mutant and control embryonic samples revealed significant changes of: (1) dPAU for 6 genes, (2) CE splicing irrespective of 3′UTR isoform (PSI_All) for 7 genes, (3) CE splicing in short 3′UTR isoform only (PSI_Short) for 5 genes, (4) CE splicing in long 3′UTR isoforms (PSI_Long) for 11 genes (Fig. 6c).

For *Khc-73*, we observed a significant reduction in dPAU in the *elav* mutant condition (Fig. 6c, e). In *elav* compared to control embryos, exon 15 PSI decreased for the long 3′UTR isoform and increased for the short 3′UTR isoform. In contrast, when measuring exon 15 PSI from all aligned reads, there was no change in *elav* mutants compared to control (Fig. 6c, e). Thus, the direction of ELAV-mediated PSI change was dependent on 3′UTR isoform. Interestingly, PL-Seq revealed that the increased retention of *Dscam1* exon 23 in *elav* mutant embryos occurred exclusively in the long 3′UTR whereas there was no change in

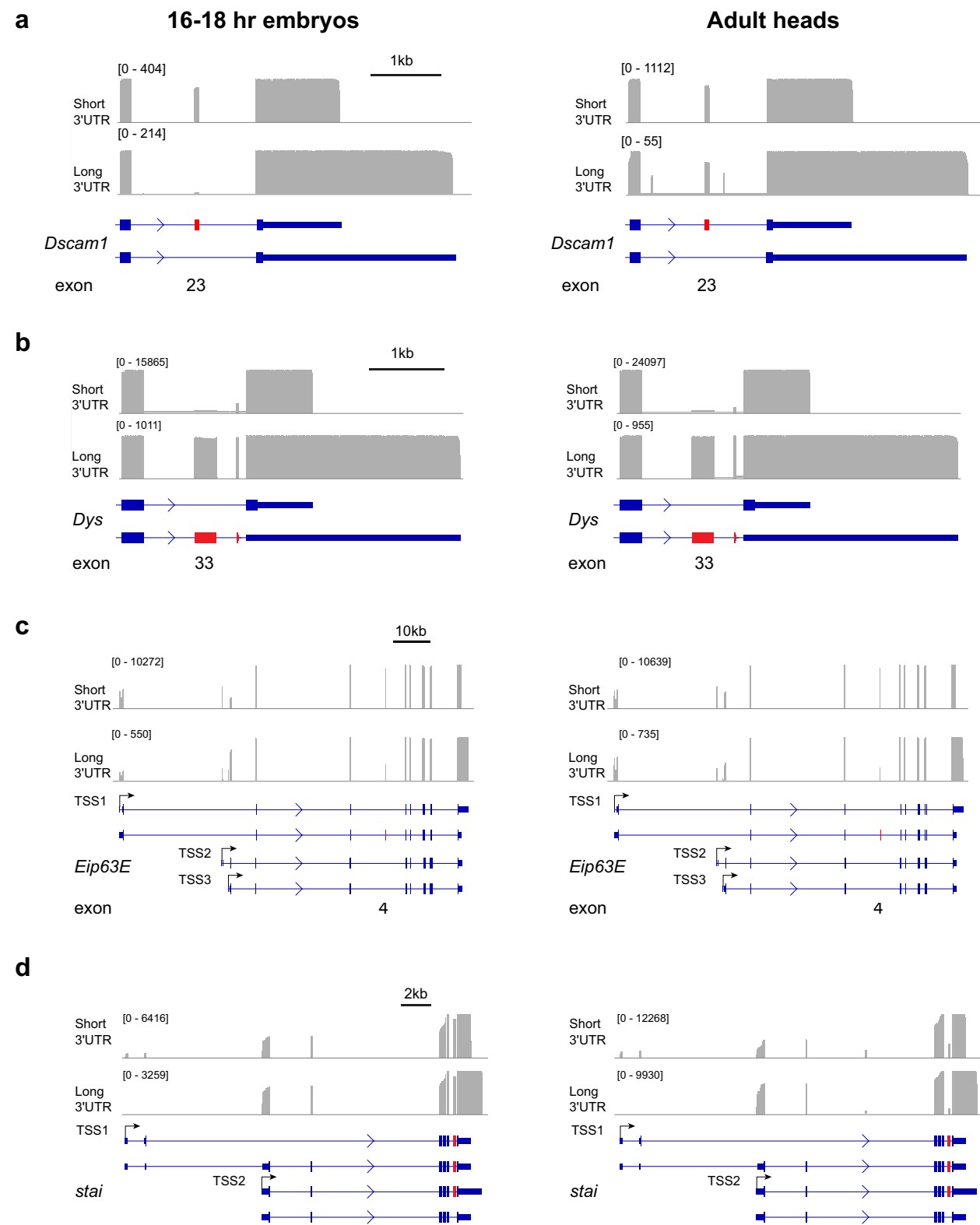

**Fig. 4 | PL-Seq tracks of genes exhibiting 3'UTR connected CE splicing.** Left side shows 16–18 hr embryos and right side shows adult heads. **a** *Dscam1*, **b** *Dys*, **c** *Eip63E*, and **d**) *stai*. For quantification of CE PSI see Fig. 3 and supplementary Fig. 5.

Note the change in alternative first exon usage for *Eip63E* and *stai* between long and short 3'UTR isoforms. Also see supplementary Fig. 6.

the short 3'UTR transcripts (Fig. 6d). This evidence from *Khc-73* and *Dscam1* suggest that 3'UTR isoform content or choice impacts ELAV-mediated alternative splicing of CEs. For both *Dscam1* and *Khc-73*, there was a significant reduction in the difference of PSI in long versus

short 3'UTR isoforms in *elav$^5$* embryos compared to control (control vs *elav$^5$* |PSI$_{Long}$-PSI$_{Short}$|, $p < 0.001$) (Fig. 6c). Remarkably, for all significant changes in |PSI$_{Long}$-PSI$_{Short}$| detected (10/23 genes), there was always a decrease in the *elav$^5$* condition (Fig. 6c). In other words, after

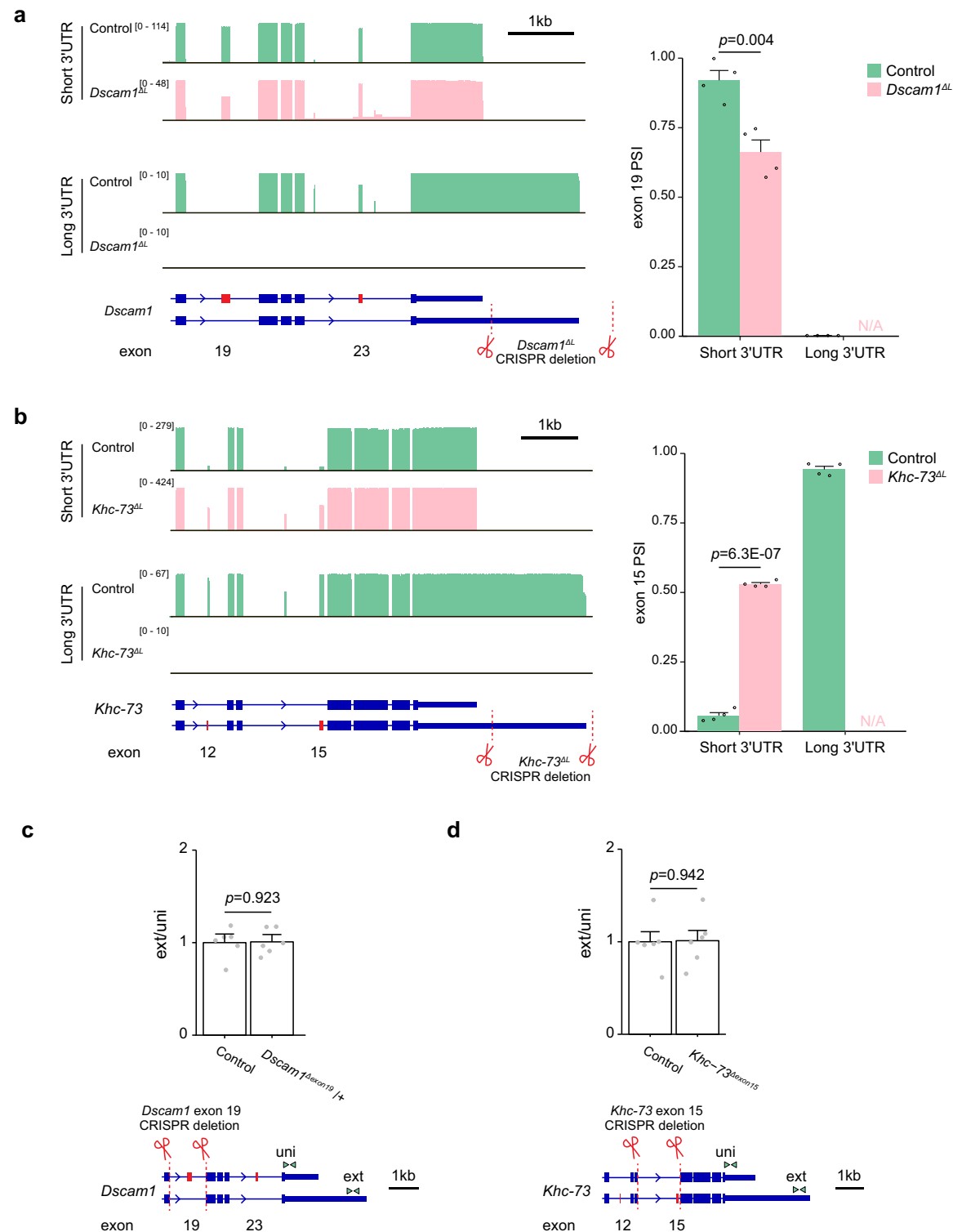

**Fig. 5 | Loss of long 3'UTR alters CE splicing in short 3'UTR isoforms. a** *Dscam1* exon 19 splicing patterns of short and long 3'UTR reads in adult heads from control and *Dscam1* long 3'UTR mutants (*Dscam1^ΔL*). **b** *Khc-73* exon 15 splicing patterns of short and long 3'UTR reads in adult heads from control and long 3'UTR mutants (*Khc-73^ΔL*). *p-value* calculated from a two-tailed t-test. **c** qRT-PCR of distal 3' polyA site usage (ext normalized to uni) in *Dscam1* exon 19 heterozygote mutants (*Dscam1^ΔExon19*/+). **d** qRT-PCR of distal 3' polyA site usage for *Khc-73* exon 15

homozygote mutants (*Khc-73^Δexon15*). Sequences deleted by CRISPR/Cas9 gene editing are illustrated by dotted red lines and scissors. The approximate location of uni and ext primers are indicated by green arrowheads. *p-value* calculated from a two-tailed t-test, *n* = 6 biologically independent samples. Data is presented as Mean + SEM. Source data are provided as a source data file.

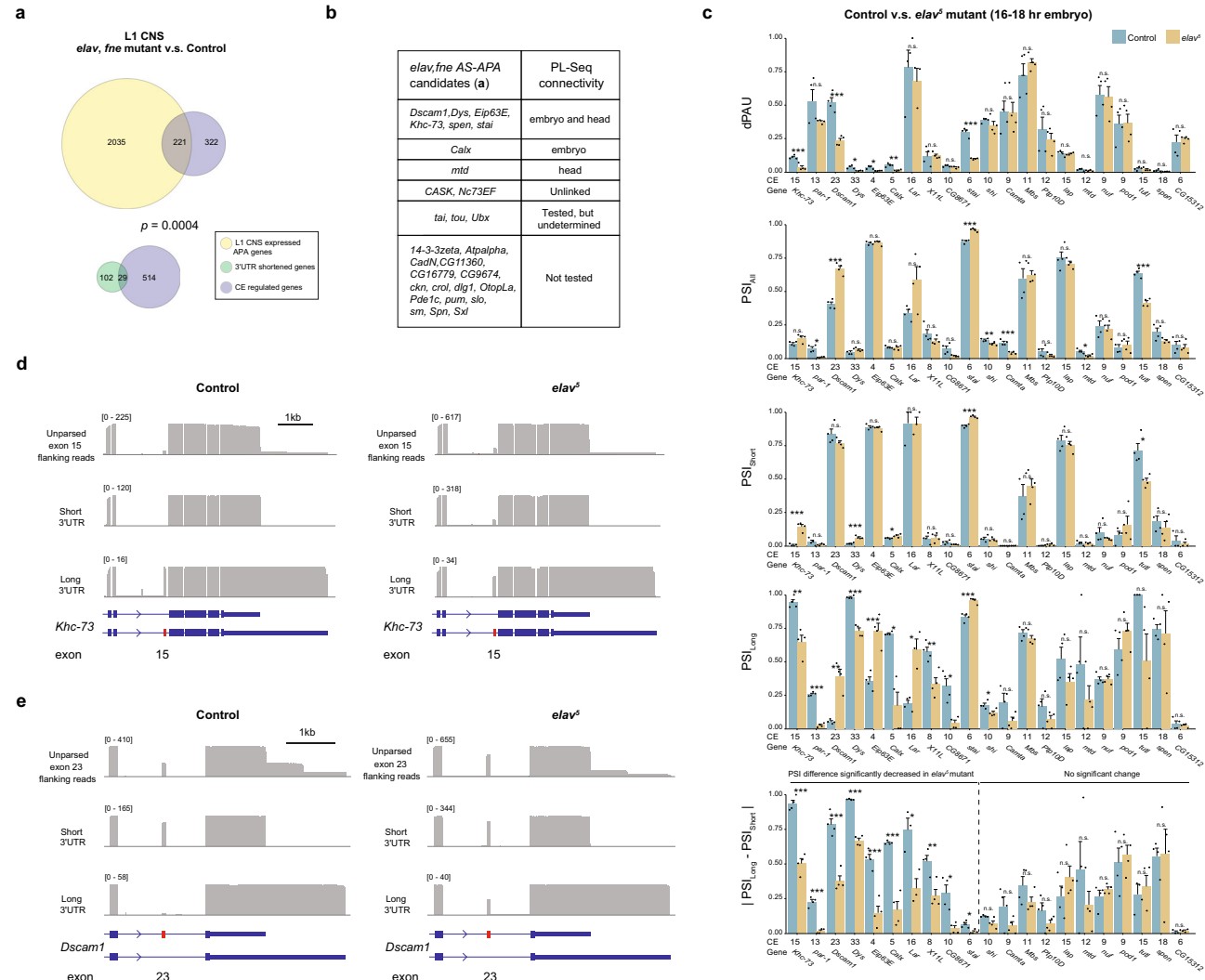

**Fig. 6 | ELAV loss of function deregulates 3'UTR connected CE splicing. a** Fisher's exact test (two-sided) shows 3'UTR shortening genes in *elav,fne* mutants are significantly associated with regulated CE events when comparing RNA-Seq data from L1 CNS *elav,fne* mutants with controls[9]. **b** List of the 29 AS and APA regulated genes in *elav,fne* mutants with PL-Seq verification information for 3'UTR connected CE splicing. **c** dPAU, PSI from all transcripts (PSI$_{All}$), short (PSI$_{Short}$) and long (PSI$_{Long}$) 3'UTR isoforms, and CE splicing difference between short and long 3'UTR isoforms (|PSI$_{Long}$ - PSI$_{Short}$|) compared between control and *elav* mutant embryos using PL-

Seq data. Data is shown as Mean + SEM. * indicates $p < 0.05$, **$p < 0.01$ and *** $p < 0.005$. Two-tailed t-test, no adjustment for multiple comparisons, n = 3-4 biologically independent samples. **d** *Khc-73* exon 15 splicing patterns of short and long 3'UTR reads in adult heads from control and *elav* mutant embryos. **e** *Dscam1* exon 23 splicing patterns of short and long 3'UTR reads in adult heads from control and *elav* mutant embryos. Also see supplementary Fig. 7. Source data are provided as a source data file.

controlling for the directionality of inclusion/skipping, the loss of ELAV tends to minimize the difference in 3'UTR mRNA isoform-specific CE PSI values. Overall, these data illustrate the utility of PL-Seq to quantify RNA binding protein-regulated changes in the intramolecular connectivity of 3'UTRs to CEs.

### PL-Seq reveals 3'UTR connected alternative splicing for the *Endov* gene in mouse ES cell-derived neurons

Neural differentiation of mouse ES cells (mESCs) has previously been found to cause overall lengthening of 3'UTRs[48]. We re-analyzed mESC neural differentiation RNA-Seq data[55] and identified 115 genes that were regulated by both 3'UTR lengthening and CE alternative splicing (Supplementary Fig. 8a, b, Supplementary Data 7). One of these genes was Endonuclease V (*Endov*), a highly conversed protein involved in DNA repair and RNA cleavage[56,57]. PL-Seq performed on mESC-derived neurons revealed the presence of three alternative length 3'UTRs for *Endov*. Parsing of reads into long, medium, and short 3'UTRs showed a significantly greater PSI of exon 4 for the long 3'UTR compared to both

the short and medium 3'UTR (Supplementary Fig. 8e, Exon 4 PSI$_{short}$ = 89.8%, Exon 4 PSI$_{medium}$ = 22.3%, Exon 4 PSI$_{long}$ = 29.8%; $p < 0.05$). These data indicate that PL-Seq can be applied to quantify connected AS-APA events in various organisms, tissues, and cell types.

## Discussion

AS and APA are key co/post-transcriptional processing events that impact most metazoan genes. Despite their importance, a limited number of studies have found evidence that alternative exon choice and 3'UTR choice are connected[18,33]. Here, we used PL-Seq, a cDNA capture-based long-read sequencing method, to investigate the interactions between AS and APA. In *Drosophila* late-stage embryos and heads, we uncovered 3'UTR connected CE splicing events for 23 genes, of which 10 were not initially recognized as potential candidates using short-read RNA-Seq analysis. These findings suggest that many more genes might be affected by connected AS-APA events. Applying PL-Seq to *elav*[5] mutants, we found CE splicing events for individual genes that were differentially regulated depending on whether reads

were connected to the short or long 3'UTR. To date, our understanding of the transcriptome-wide AS events regulated by RBPs has largely been based on short-read data. Long-read sequencing might uncover a hidden layer of RBP-regulated AS events that can only be detected when connectivity to 3'UTRs is considered.

To obtain a broader scope of 3'UTR linkage to CE splicing, PL-Seq could be applied to all 694 genes expressed in *Drosophila* embryos/heads that are annotated for both CEs and alternative length 3'UTRs. Our analysis of short-read RNA-Seq data from mouse ES neuronal differentiation and PL-Seq performed for *Endov* also suggests that a larger scale investigation could uncover many connected AS-APA events during mammalian neuronal differentiation (Supplementary Fig. 8). Outside of the nervous system, there are surely other connected AS-APA events waiting to be discovered in different tissues, developmental time points, disease states, and cell types exhibiting regulation of APA[1,58]. We restricted our sequencing to explore CE alternative splicing, but there is likely a plethora of exon to 3' end connectivity events that await discovery with long-read sequencing. Very recently, extensive connectivity between alternative first exons and alternative 3'UTRs was established using transcriptome-wide long read sequencing, including for *stai* and *Eip63E* which we confirm here by PL-Seq[46] (Fig. 4c, d). Functional experiments supported a role for alternative promoters in driving 3'UTR choice for several genes as evidenced by genomic promoter deletion and CRISPR activation of alternative promoters[46]. Regulation of APA clearly involves more than the binding of RBPs and the core cleavage and polyadenylation machinery in the vicinity of polyA sites– roles for enhancer/transcription activity, DNA methylation, and specific chromatin remodeling proteins have emerged in recent years[28,59–62]. Given this widening landscape of regulatory influences, future studies on the mechanisms of APA cross-talk with other co-transcriptional events will need to employ long read sequencing.

cDNA capture-based nanopore sequencing methods such as PL-Seq are valuable additions to the transcriptomics tool-box[44,45]. As long read sequencing continues to advance, constraints related to read length and depth that currently limit their transcriptome-wide application will likely be resolved. Until then, pull-down-based approaches such as PL-Seq offer a valuable means of quantifying the exon composition of alternative 3'UTR isoforms. PL-Seq is low-cost and requires little to no capital investment. The method and analysis pipeline can be applied to quantify exon to 3'UTR connectivity for specific genes of interest by any laboratory equipped for molecular biology research. While we used a limited number of probes and targets in our current work, capture-based cDNA pulldown is effective at enriching thousands of targets simultaneously and can be scaled up accordingly[42,43,45]. PL-Seq should also be adaptable to single-cell analysis, providing a targeted approach to complement recent advances in long-read single cell RNA-seq[33,63]. Methods that can accurately quantify the specific exonic composition of full-length transcripts at the single-cell level will be crucial for understanding how regulation of APA and other co-transcriptional events are coordinated with each other in complex tissues such as the brain.

Tandem 3'UTR APA events, by definition, do not alter the protein-coding potential of the mRNA isoforms produced, in contrast to intronic APA or alternative last exon APA. However, here we identify genes with extreme connectivity of upstream CEs to tandem 3'UTRs. In these cases, tandem 3'UTR APA choices are inseparable from the production of different protein isoforms (e.g. *Dys*, *Khc-73*, *Dscam1*). Such examples do not fit neatly into our current classification systems of AS and APA. In most of the 23 AS and APA-connected genes we study here, the differences between the isoforms are only in the sequence content of the alternative exons. For other cases, CE inclusion can cause a frameshift that changes the C-terminus of the protein more drastically. For instance, the inclusion of CEs in the *Dys* long 3'UTR isoform leads to the stop codon shifting to the second last exon and

eliminates the protein-coding capacity of the terminal exon (Fig. 4b). These findings, along with the strong correlation between alternative promoter selection and 3'UTR APA (which often modifies protein-coding exon content)[46], suggest that it is no longer appropriate to assume that tandem 3'UTR APA events generate mRNAs that differ solely in their non-coding sequence composition.

To explore the mechanism of intramolecularly connected AS and APA, we generated CRISPR deletion mutants lacking either long 3'UTR isoforms or inclusion isoforms of specific CEs. Previous studies have shown that splicing can affect 3' end processing[36,64,65]. While we observed that the loss of upstream CEs and their adjacent introns did not impact polyA site selection in the case of *Dscam1* and *Khc-73*, this does not eliminate the possibility that the connection between exons and 3'UTRs observed in other genes could be influenced by co-transcriptional alternative splicing events. The mechanism of how loss of long 3'UTR alters CE splicing in the short 3'UTR isoforms for *Dscam1* and *Khc-73* remains unclear. The short and long 3'UTRs of these genes might impart different RNA stabilities via microRNA and RBP target sites in the 3'UTR or via differences in polyA tail length[66]. Differences in expression patterns of these mRNA isoforms with regard to cell type and subcellular localization might also play a role. Our study quantifies transcript isoforms at steady-state levels. The future investigation that takes into account the timing of 3'UTR transcription and processing relative to splicing in vivo, using metabolic labelling and nascent RNA technologies coupled to long-read sequencing[36,38], might lend new insights into how these RNA processing events are coordinated on individual genes.

## Methods

### *Drosophila* CRISPR/Cas9 deletion lines

Fly CRISPR/Cas9 genome editing was performed by WellGenetics Inc. A homology-directed repair strategy was utilized to generate flies harboring a deletion of the *Khc-73* long 3'UTR sequence (*Khc-73^ΔL*). Briefly, two gRNAs were designed targeting the long 3'UTR to remove the genomic region spanning chr2R:15515657-15517432. The donor plasmid containing the homologous arms with the deletion and two loxP sites bracketed 3xP3-RFP cassette was injected into embryos of control strains together with targeting gRNAs. Flies carrying RFP markers were selected and further validated by genomic PCR and sequencing. Validated mutants were crossed to flies expressing Cre recombinase to remove the 3xP3-RFP insertion.

For CE mutants, exons of interest and their flanking introns were deleted using CRISPR and homology-directed repair. 3xP3-DsRed flanked by PiggyBac terminal repeats was embedded in a TIAA motif in the homologous sequences to avoid additional restriction digestion residues. For the *Khc-73* exon 15 mutant, the genomic region spanning chr2R:15520337-15521841 was deleted (*Khc-73^ΔExon15*). For the *Dscam1* exon 19 mutant, the genomic region spanning chr2R:7323333-7324569 was deleted (*Dscam1^ΔExon19*). DsRed was used as the screening marker and then excised from validated mutant flies by crossing to flies expressing the PiggyBac transposase. *w^1118* was used as the wild-type strain, and all deletion strains were generated from *w^1118* background.

### mESC and glutamatergic neuron differentiation

mESC (E14TG2a) cells between passages number 5–15 were routinely maintained in mESC medium (DMEM supplemented with 1x Glutamax, FBS, β-mercaptoethanol, MEM non-essential amino acids, sodium pyruvate, and LIF) on MEF feeder or on gelatin-coated tissue culture dishes. mESC media was replaced daily and split every other day. mESC differentiation is performed following this protocol: $3.5 \times 10^6$ cells were plated onto 90 mm bacteriological dishes in 15 mL NPC medium (DMEM with L-glutamine (Thermo Scientific 11965092) supplemented with 10% FBS, 1X non-essential amino acids, and 550 μM β-mercaptoethanol). Media change was performed on day 2 (equivalent of DIV −8). On day 4, 6, and 8, media change was performed and 5 μM of

retinoic acid was added. On day 10, NPC aggregates were collected and dissociated with 1 mL of TrypLE (Thermo Scientific 12604013) at 37 °C for 5-7 min. To halt the reaction, 8 mL of Trypsin inhibitor (Thermo Scientific R007100) was added. The NPC aggregates were gently dissociated by pipetting up and down and filtered through 40 µm cell strainer. Cell suspension was diluted in N2 media (Neurobasal (Thermo Scientific 21103049) supplemented with 1X N2 (Thermo Scientific 17502048) and 2 mM glutamine (Thermo Scientific 25030081)) at 3 × $10^5$ cells/mL. 10 mL of cells were plated onto PDL (Sigma P7280) coated 100 mm cell culture dishes. Complete media change was performed at 4 h (day 10) and 24 h (day 11) with N2 media. On day 12 (equivalent of DIV 2) and 14 (DIV 4), media was replaced with B27 media (Neurobasal supplemented with 1X B27 (Thermo Scientific 17504044) and 2 mM glutamine). Cells were maintained until day 17 (DIV 7) and then collected for RNA extraction.

### Short-read RNA-Seq data-based alternative splicing and alternative polyadenylation analysis

For AS and APA analysis, we used publicly available RNA-Seq data from fly tissues under BioProject accession PRJNA75285[47]. For AS analysis, RNA-Seq reads were aligned to the *Drosophila Melanogaster* (dm6) or *Mus musculus* (mm10) genome using STAR 2.7. Sorted output bam files were then fed into rMATS (v4.0.2)[50] to identify alternatively spliced CEs. The output file was filtered by FDR < 0.05 and |IncLevelDiff| > 0.2 to create a gene list of differential spliced events of high confidence. For APA analysis, QAPA 1.2.3 was used with Ensembl gene dm6 3′UTR annotations. Custom R scripts were used for filtering as follows: only genes with gene-level TPM values greater than 0 were considered as expressed and thus included in the downstream analysis; dPAU values were set by filtering the maximum length of the 3′UTR sequence detected by QAPA; fold change of dPAU values between samples was calculated and two-tailed t-test was performed followed by FDR correction; genes that were differentially regulated at both AS and APA levels were identified (fold change of dPAU>2 or <0.5 plus FDR < 0.05), and their association was tested by two-sided Fisher's exact test. QAPA and rMATS analysis tables can be found in Supplementary Data 1, 2, 4, 6 and 7. Gene Ontology analysis was performed on FlyEnrichr[67]. Data arrangement, statistical analysis, and graph generation was performed in R 3.6.1 and R 4.2.2.

### RNA extraction, cDNA synthesis, nested RT-PCR and qRT-PCR

Fly embryos from various time points and adult heads from mixed 1–5-day-old males and females were collected. Total RNA was extracted using Trizol (Thermo Fisher Scientific) per the manufacturer's instructions. Briefly, samples were triturated and lysed in Trizol on ice. Phase separation was performed by adding 1/5 volume of chloroform and centrifuging at 20,000 x *g* for 20 min at 4 °C. Upper aqueous phase was precipitated with isopropanol and centrifuged at 20,000 x *g* for 20 min at 4 °C. RNA pellet was washed with 70% ethanol (ethanol was removed after centrifugation at 20,000 x *g* for 10 min at 4 °C), and then resuspended in desired volume of distilled water. RNA was quantified using a Nanodrop spectrometer.

For cDNA synthesis, 1 µg of Turbo DNase (Thermo Fisher Scientific) treated total RNA was reverse transcribed using Maxima reverse transcriptase (Thermo Fisher Scientific). For RT-PCR of uni, ext, and alternative splicing events (Supplementary Fig. 1) PCR was performed using Taq DNA polymerase with standard buffer (NEB). PCR products were resolved in agarose gels and imaged using Gel Doc EZ (Bio-Rad). Exposure time was adjusted to ensure band intensities were not saturated. PSI values were estimated using a gel analyzer tool in Image Lab Software (Bio-Rad). For qRT-PCR analysis, 2 µL of 1:10 diluted cDNA (in water) was used as the template, 1 µL of each primer (10 µM), 10 µL of SYBR Select Master Mix (Thermo Fisher Scientific), and 7 µL water was added to each reaction. Samples were subjected to 40 amplification cycles, and data was collected and analyzed using the delta delta Ct

method on CFX Maestro Software (Bio-Rad). Primer sequences can be found in Supplementary Data 8.

### PCR-based gene-specific nanopore sequencing

For PCR based Nanopore library preparation, PCR amplicons using *Khc-73* specific primers with barcoding adapter sequences were used. Each sample was barcoded by PCR using Nanopore PCR barcoding kit (EXP-PBC001). Barcoded samples were pooled at equimolar concentration, and end-prepped using NEBNext FFPE DNA Repair Mix and NEBNext Ultra II End Repair Kit. The nanopore adapter was ligated using Nanopore ligation sequencing kit (SQK-LSK109). Alternatively, samples were prepared without barcoding and sequenced separately. In this case, PCR amplicons at the equimolar concentration were end-prepped and the nanopore adapter was directly ligated. MinION Mk1B device and FLO-FLG001 flow cells were used for sequencing of the libraries.

### PL-Seq library preparation

For full-length cDNA synthesis, SMARTer PCR cDNA synthesis kit (Clontech) was used according to the manufacturer's specifications. Total RNA was Dnase treated on-column using PureLink Dnase (Thermo Fisher Scientific) and the PureLink RNA Mini kit (Thermo Fisher Scientific). First-strand synthesis was performed using ~500 ng of Dnase treated RNA, 3′ SMART CDS Primer II A, and SMARTer II A TSO. cDNA was diluted 1:5 in TE buffer, and then used as the template to synthesize double strand cDNA amplicons by PCR (optimal 17 – 21 cycles) using Advantage 2 PCR kit according to the manufacturer's specifications (Clontech). cDNA was purified using NucleoSpin Gel and PCR Clean-up kit (Takara Bio). cDNAs of our interest were enriched by pulldown starting with 5-10 µg of PCR amplified cDNA and custom designed 5′ biotinylated oligonucleotide xGen Lockdown probes (Integrated DNA Technologies) and the xGen hybridization and wash kit (Integrated DNA Technologies). Probe sequences are listed in Supplementary Data 8. Captured cDNA was amplified using Takara LA Taq DNA polymerase Hot-Start version (Clontech) and purified using 1:1 (vol: vol) AMPure XL beads (Beckman Coulter). cDNA prepared as above was end-prepped using NEBNext Companion Module for Oxford Nanopore Technologies Ligation Sequencing (NEB) and the nanopore adapter was ligated using Nanopore ligation sequencing kit (SQK-LSK110). Thirty µL of the prepared library was then loaded onto a flow cell (FLO-FLG001) and sequenced using Nanopore MinION Mk1B sequencer.

### PL-Seq data read mapping, filtering, and analysis

Reads were aligned to fly genome assembly (dm6)/mouse genome assembly (mm39) and transcriptome using minimap2 (v2.17) with the arguments -ax splice -B 3 -O 3,20 to allow optimized splice junction recognition. Aligned files from minimap2 were converted to bam format using SAMtools (v1.6)[68] and then a quality check was performed using tools in NanoPack (v1.41.0)[69] and coverage was examined using RSeQC (v5.0.1)[70,71]. Reads aligned to targeted genes were counted by featureCounts[72]. For downstream analysis, aligned reads were subjected a series of sequential filtering using a custom python script exon_coverage.py[73]. This allows for only including reads that cover upstream constitutive exons and downstream universal 3′UTR regions with subsequent parsing for short and long 3′UTR isoforms. The 3′ ends were inferred by drops in Nanopore read coverage in the 3′UTR region and current Ensembl genes annotations. Reads identified as short 3′UTR isoform-specific were fed to another custom python script polyA_filtering.py to filter out truncated reads lacking the polyA tail in the soft clipped region and reads that are internally misprimed due to genomically encoded A enriched sequences in the transcripts. Two minimum read counts were required for PSI calculation. PSI values of cassette exons corresponding to short or long 3′UTR isoforms were generated using a third script, calculate_PSI.py and tested by pairwise

t-test with 3 or more replicates. When 3 or more tandem polyA sites were used in the gene, long was defined as the most distal 3'UTR and short as the most proximal one. Details for the 3'UTR connected CE we used for the analysis can be found in Supplementary Data 9. Read count summaries corresponding to short and long 3'UTR isoforms for all PL-Seq experiments is summarized in Supplementary Data 10.

### Reporting summary
Further information on research design is available in the Nature Portfolio Reporting Summary linked to this article.

## Data availability
All sequencing data presented in this study have been deposited at the Sequence Read Archive (SRA) with Bioproject accession number PRJNA771049 [https://www.ncbi.nlm.nih.gov/bioproject/PRJNA771049/]. The short read RNA-Seq datasets analyzed included *Drosophila* tissues and embryos with the accession number PRJNA75285 [https://www.ncbi.nlm.nih.gov/bioproject/?term=PRJNA75285], *Drosophila elav/fne* mutant tissues with the GEO series number GSE155534 [https://www.ncbi.nlm.nih.gov/geo/query/acc.cgi?acc=GSE155534], and mESC neural differentiation with the accession number SRP017778 [https://www.ncbi.nlm.nih.gov/sra/?term=SRP017778]. Source data are provided with this paper.

## Code availability
The custom scripts for PL-Seq workflow are available from https://github.com/markandtwin/Pull-a-long and at the online repository zenodo.org with the accession code 8215376 [https://doi.org/10.5281/zenodo.8215376][73].

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

## Acknowledgements

We thank Dr. Christopher Vollmers (UC Santa Cruz) for Nanopore sequencing methodology discussions and Dr. Jung Hwan Kim (University of Nevada, Reno) for insights and discussion on the manuscript. Thanks to Miura lab members for reading and providing input on the manuscript. This work was supported by NSF IOS grant 1656463 and NIGMS grant R35GM138319 awarded to P.M. Core facilities at the University of Nevada, Reno campus were supported by NIGMS COBRE P30GM103650.

## Author contributions

Conceptualization, Z.Z., W.H.D., B.B., and P.M; Methodology, Z.Z., W.H.D., B.B., and P.M.; Investigation, Z.Z., B.B and W.H.D.; Data Analysis, Z.Z., B.B, W.H.D., and P.M. Software, Z.Z.; Writing, Z.Z., and P.M.; Funding Acquisition, P.M.; Supervision, P.M.

## Competing interests

The authors declare no competing interests.
