## [Peer Review File · Nature Communications]

Coordination of Alternative Splicing and Alternative Polyadenylation revealed by Targeted Long Read SequencingREVIEWER COMMENTS

Reviewer #1 (Remarks to the Author):

This study by Zhang, Bae, Cuddleston, and Miura explores the link between upstream cassette-exon (CE) alternative splicing and alternative polyadenylation. They show that alternative CE AS occurs in 45% of genes that undergo ELAV-dependent 3' UTR lengthening. By combining cDNA pull-downs with nanopore long read sequencing (LRS) in an approach they name PL-seq, the authors demonstrate that in 23 genes, CE alternative splicing is directly connected to alternative polyadenylation, i.e., alternative exons and alternative 3' UTRs occur in the same RNA molecule. Finally, the authors show the concomitant loss of ELAV-dependent CE and 3' UTRs for a number of genes in *elav* mutant embryos.

The global loss of neuron-specific CEs and 3' UTRs in *elav* mutant embryos has been demonstrated before (Carrasco et al., 2020; Wei et al., 2020; Lee et al., 2021). The association of AS and APA events in ELAV-dependent genes was also known: shown in *Dscam1* (by the same lab in a 2019 publication), and at a global scale by others (Carrasco et al., 2020, Lee et al., 2021). Moreover, the use of cDNA capture coupled to LRS has been described before, for example ORF Capture-Seq (Ray et al., 2020, Sheynkman et al., 2020, Singh et al., 2019, Kovaka et al., 2020). The long-read validation of alternative splicing and alternative polyadenylation in >20 genes is very interesting, but was not followed up by mechanistic or functional studies. Therefore, the gain in conceptual insight or technological novelty that this study provides, is relatively limited.

Specific comments:

- In Figure 1 as well as S1, and Fig. 6, according to the Venn diagrams, 21-23% of 3'UTR lengthened genes undergo CE regulation, which greatly differs from the 50% stated elsewhere. The 50% refer to the very small set (ten) that were experimentally validated by LRS, and therefore are not representative for a global claim. Besides, it was stated before that about one-third of neurally lengthened genes undergo AS (Carrasco et al., 2020), which the authors do not mention. Hence, this important claim lacks transparency as well as novelty.
- Figure 3 results: More statistics on the efficiency of PL-seq in pulling down full-length isoforms should be provided. For example: transcript coverage metrics as addressed in Soneson et al., 2019; read length before and after filtering; junction coverage; % of splice signals and exon-intron junctions.
- Figure 3: The approach the authors use to show exon inclusion within 3' UTR isoforms in the parsed long reads, could lead to confusion. For example, the total rate (in absolute terms) of exon production could still be higher for the short 3' UTR isoform. Authors show tracks of different scales; this enhances the view for specific exons that may not be necessarily be coupled to the 3'UTR, and should be avoided.
- For an accurate representation of the exon-3'UTR link, it would be helpful to compare the total abundance of a given exon to the overall gene expression.
- Figure 6 results: from the absence of an exon from the short, but not the long 3' UTR isoform in *Dscam1* mRNAs, the authors conclude that "3'UTR isoform content or choice impacts Elav-mediated alternative splicing of exon 23". I disagree with this interpretation. The results show that the two events are linked, but not how. The causation could be the other way around, i.e., alternative splicing leads to APA, or, perhaps the most plausible explanation, ELAV binding causes both AS and APA on the same transcript.
- A de novo assembly of ELAV-dependent isoforms found only in the mutant would add more interesting insights of isoform regulation and ELAV-dependent 3'UTR choice.
- The *elav5* molecular study lacks quality controls that support ELAV and control datasets are of good quality and comparable.
- More insights into the regulation would have been desirable, for example by complementing this analysis with molecular studies (for example, ELAV CLIP and iCLIP studies are available) or functional studies (deletion of one or more CE to assess effect on associated 3'UTR).

Minor comments:

- Figure 1 results: To identify neuronal alternative CE AS, much better datasets than those used by the authors are now available: in sorted neurons and other cell populations (McCorkindale et al., 2019), and in elav mutant neuronal tissues (Wei et al., 2021). Figure 1 is overall redundant with previous studies.
- The abbreviation CE for cassette exon is not explicitly stated
- Typos are found throughout the text ("publically" at beginning of results section; and others).
- The way the results are presented could be improved to increase readability. For example, in Fig. 4c, numbers/proportions/ratios should be provided instead of tables with yes/no.

Reviewer #2 (Remarks to the Author):

In their manuscript Zhang and co-workers essentially report a novel method ('PL-Seq') to elucidate the link between alternative splicing and 3'UTR isoforms. Using this probe-based cDNA pulldown approach coupled to Nanopore sequencing, the authors identify more than 20 genes that exhibit alternative splicing linked to 3'UTR isoforms in neuron-enriched Drosophila tissues. They also show that the neuron-specific RNA-binding protein ELAV plays a role in this coordinated AS-APA regulation in Drosophila.

The variant of direct long-read sequencing introduced here is interesting and versatile. It helps to illuminate the landscape of co-regulated RNA processing and to decipher underlying mechanisms. The paper also provides a script on github for the PL-Seq workflow.

While I see a clear methodological advance, the underlying mechanisms and resulting biology of the coupling between AS and APA, which is indeed striking, remain somewhat vague.

Specific points:

- What is the function of the transcripts co-regulated by AS and APA? Where do such RNAs localize in neuronal cells? Any obvious biological function that can be aligned to this type of regulation inferred from sequence, encoded protein domains etc? Does the 'dual specification' regulate spatially restricted protein functions in neuronal cells?
- Overall, the number of coupled AS-APA events seems surprisingly small (given the enormous diversification of transcripts in Drosophila such as DSCAM etc). Is this limited extent specific to Drosophila? How does this compare to vertebrates?
- The abstract seems ambivalent: Is the focus on the method or the biology?

Minor:

- Introduction 1st paragraph, result section 1st paragraph: Recently, PCF11 was shown to be a quintessential regulator in neuronal development by regulating 3'UTR isoforms (PMID: 30552333). Is there evidence that PCF11 plays a role in the model used here?
- Introduction 1st paragraph: The translational significance of 3'UTR regulation could be strengthened (i.e. disease relevance)
- Introduction 3rd paragraph, result section 1st paragraph: Acronym 'CE' page 2 and 3 not explained
- Supplementary Figure 1: legend title missing
- Abstract 2nd sentence: remainder of a previous reference

Reviewer #3 (Remarks to the Author):

This timely manuscript investigates linked RNA processing events, namely coordinated alternative splicing (AS) with alternative polyA site usage (APA). The study follows up a previous report by the same authors (Zhang, Z. et al. 2019. Elav-Mediated Exon Skipping and Alternative Polyadenylation of the Dscam1 Gene Are Required for Axon Outgrowth. *Cell Rep* 27, 3808-3817.e3807, doi:10.1016/j.celrep.2019.05.083) showing that changes in Dscam alternative splicing and APA as development proceeds is coordinated by the RNA binding protein ELAV, which is also investigated here. A series of papers from the Hilgers lab on these topics previously linked APA regulation by ELAV to transcriptional pausing, also in the fly nervous system. The major advance comprised in this paper is a targeted survey of AS and APA events in 31 genes using long read sequencing, a third generation RNA-seq method that enables detection of AS and APA status in the same long read. Despite the rigor of some of the data and their analyses, the present manuscript lacks biological significance and mechanistic insight. For example, the developmental setting is not exploited except as a source for library preparation. Given the previously published body work, the major gap in knowledge would seem to be the generality and/or mechanistic features that coordinate AS and APA choices. The manuscript does not do this. We make the following suggestions:

Major points

1. The findings lack novelty and mechanism

The conceptual framework ignores obvious mechanistic features that could be tested. The authors completely avoid the question of whether APA site choice determines AS, or if it is the other way around. Most readers will know that 3' end cleavage of mRNA at the polyA site is, by definition, a cotranscriptional event that initiates transcription termination. They will also know that most introns are spliced co-transcriptionally in many species, including fly (papers by Rosbash and Carmo-Fonseca). Analysis of nascent RNA, rather than polyA+ mRNA, would answer the question of whether AS occurs before 3' end cleavage. If the 3'UTR affects upstream splicing, then these splicing events must be post-transcriptional; this would be an interesting and novel finding that would reflect a mechanistic feature. Previous studies also indicate coordination between splicing and 3' end cleavage, which are in some cases mediated by specific splicing regulators. Finally, generality of the reported coordination between AS and APA also seems to be lacking. Figures 1E and 6C indicate that many APA events are not coordinated with AS and vice versa. In this regard, figure 6F is not very helpful.

2. Methods and Data analysis.

Because the authors did not achieve a depth of coverage that allowed detailed analysis of the desired genes, they adapted the library prep to include enrichment of the cDNAs of interest using biotinylated probes; following this step, the captured cDNAs are reamplified and subjected to final library preparation and MinION sequencing. They dub this method Pull-a-Long-Seq (PL-seq). PL-seq is easier to multiplex than PCR, so we would consider it an advantageous variation on targeted long read sequencing. Given this, the methods and data analyses must be crystal clear. They state, "Two to five biotinylated probes were designed to target constitutive exons and universal 3'UTR sequences for each targeted gene". This is a far lower density than traditional Capture-seq, but what is the density of probes? It is also unclear if they are only targeting one strand or both of each cDNA. What is the efficiency of targeting in terms of yield? Does this procedure introduced in the middle of the protocol introducing biases? It seems like it would be possible to address this by comparing long reads from a highly expressed gene that are not enriched versus enriched.

We would like to see some of the actual reads and not just a pile up of reads. We find it odd that the data shown (e.g. Fig 2C) are so block-like. Long read sequencing usually includes a lot of incomplete reads. Have these been filtered out? Also, the data processing section is inadequate. What are the minimap parameters? Did they filter out PCR duplicates?

3. Figure legends do not describe the data panels and need to be improved. The text currently provided is very interpretive, telling us what the data show rather than what the data are. Some of the figures themselves could be improved. Figure 2A is too simplistic and could be removed; the use of the word "linked" is a stretch because these are not necessarily functional links but correlations. Figure 3C, please indicate that this is a zoom in of the gene shown in B. Otherwise, it is confusing that the shape of the read density is different.

On the bottom of the introduction page, it is stated: "Here we find that 80% of genes examined exhibit 3'UTR linked AS in neuron-enriched tissues". This compelling metric is never mentioned again. Where does this come from? Is it globally analyzed genes? Or is this from the targeted 31 genes? If targeted, then the 80% statement is a bit misleading; 31 cherry-picked genes do not invite generalization.

Minor points:

Bottom third of the first page of the results: Fig 1C is about APA. These data are also in D but the numbers do not match up. Is the citation of the figure incorrect and there is another figure somewhere?

Typos: "upstream constitute exon"

Fig 1B: there is a green box around one line of data; what is that?

Figure 4C contains the same data and A+B. We suggest moving C to the supplement, since significance is only indicated with "Yes" or "no". The data in A&B are more quantitative. Some of the examples from Fig 5 could be put into Fig 4. It is not necessary to have the extra figure.

Figures are not correctly referenced in the text (i.e. in-text references do not point to the correct data).

Response to Editorial Assessment: Thank you for the editorial assessment and the expert reviewers' comments. We have addressed all the questions and comments posed by performing new experiments and revising the text. New data and analysis are presented in Fig. 1f, Fig. 2c, Fig. 3a-d, Fig. 5a-d, Fig. 6c, Fig. S4, S5. As suggested by reviewers, new transgenic fly lines were generated to disrupt cassette exon splicing and long 3'UTR biogenesis to investigate the communication between cassette exons and 3'UTRs (Fig. 5). The manuscript is more focused now on the novel regulatory events that PL-Seq can detect—namely changes in cassette exon alternative splicing that occur differentially depending on connectivity to long or short 3'UTRs. We have clarified details and performed new analysis to demonstrate the effectiveness of the PL-Seq technique.

We have been able to address the question of directionality of communication between alternative splicing and APA events. We show that deleting long 3'UTRs for *Dscam1* and *Khc-73* impact the splicing patterns of cassette exons far upstream for the remaining short 3'UTR transcripts by PL-Seq. In the opposite scenario—deletion the cassette exons—we find that APA is not altered (Fig. 5).

We hope the reviewers feel that this manuscript will be of wide interest to the readers of *Nature Communications* given the establishment of a method and analysis pipeline that is used to discover that cassette exon splicing regulation can be differential depending on 3'UTR content. Our paper provides concrete evidence that regulation of individual RNA biogenesis events should be considered not as isolated events, but with regard to the context of the full-length mRNAs they form. We show for multiple genes that exon choices are combined with different 3'UTRs in a non-random manner.

Please see our point-by-point responses to the reviewers' comments below. Note that we have added numbering to the reviewer comments.

REVIEWER COMMENTS

Reviewer #1 (Remarks to the Author):

R1-1:

This study by Zhang, Bae, Cuddleston, and Miura explores the link between upstream cassette-exon (CE) alternative splicing and alternative polyadenylation. They show that alternative CE AS occurs in 45% of genes that undergo ELAV- dependent 3'UTR lengthening. By combining cDNA pull-downs with nanopore long read sequencing (LRS) in an approach they name PL-seq, the authors demonstrate that in 23 genes, CE alternative splicing is directly connected to alternative polyadenylation, i.e., alternative exons and alternative 3'UTRs occur in the same RNA molecule. Finally, the authors show the concomitant loss of ELAV-dependent CE and 3'UTRs for a number of genes in elav mutant embryos.

The global loss of neuron-specific CEs and 3'UTRs in elav mutant embryos has been demonstrated before (Carrasco et al., 2020; Wei et al., 2020; Lee et al., 2021). The

association of AS and APA events in ELAV-dependent genes was also known: shown in Dscam1 (by the same lab in a 2019 publication), and at a global scale by others (Carrasco et al., 2020, Lee et al., 2021). Moreover, the use of cDNA capture coupled to LRS has been described before, for example ORF Capture-Seq (Ray et al., 2020, Sheynkman et al., 2020, Singh et al., 2019, Kovaka et al., 2020). The long-read validation of alternative splicing and alternative polyadenylation in >20 genes is very interesting, but was not followed up by mechanistic or functional studies. Therefore, the gain in conceptual insight or technological novelty that this study provides, is relatively limited.

Thank you very much for the comments. We agree with the reviewer that Elav mediated AS and APA changes have been demonstrated by the Hilgers and Lai groups using short read RNA-Seq analysis. We add here that the overlap of Elav regulated AS and APA on a per gene basis is statistically significant. The major difference with our manuscript is that we are able to quantify the connectivity of these events in individual genes using long-read sequencing. We show that PL-Seq is able to quantify CE alternative splicing differences that differ depending on whether they are connected to short or long 3'UTRs.

We agree that our method is similar in approach to previously described capture-based methods. We apologize for not extensively citing these and other prior works and we do so now. These previous works, however, have not leveraged capture-based long read sequencing for analyzing connectivity of alternative exons to alternative 3' ends. We would like to point out that our approach also includes a bioinformatics pipeline for distinguishing CE PSI calculations according to short versus long 3'UTR isoforms. The major advance here is that by applying our sequencing approach to a specific set of genes— those that are regulated by alternative splicing of Cassette Exons and APA— we are able to uncover that CE PSI can be widely different depending on which 3'UTR it is connected to.

Regarding the lack of mechanistic or functional studies, we add new data that addresses the question whether AS can affect APA, and the converse, whether loss of long 3'UTR can affect AS. By generating and analyzing 3 new CRISPR deletion lines, we show that preventing CE alternative splicing does not impact APA for *Dscam1* and *Khc-73*. In contrast, deleting long 3'UTR for *Dscam1* and *Khc-73* causes alteration in the PSI of upstream CEs (Figure 5). We explain these results further below in the answers to specific comments.

Specific comments:

R1-2:

In Figure 1 as well as S1, and Fig. 6, according to the Venn diagrams, 21-23% of 3'UTR lengthened genes undergo CE regulation, which greatly differs from the 50% stated elsewhere. The 50% refer to the very small set (ten) that were experimentally

validated by LRS, and therefore are not representative for a global claim. Besides, it was stated before that about one-third of neurally lengthened genes undergo AS (Carrasco et al., 2020), which the authors do not mention. Hence, this important claim lacks transparency as well as novelty.

Thank you for pointing this out. We agree that naming these percentages which refer to a small subset of genes can be misconstrued as being representative of global percentages. We have removed these statements. The 50% previously referred to the 10/21 genes analyzed by PL-Seq analysis found to show reduced 3'UTR dependent CE splicing difference in the *elav* mutant. We have thus changed the text to state “10 genes” instead of 50%. See also comment **R3-6**.

Regarding the statement of “roughly one third of ELAV AS targets also undergo APA” (Carrasco et al., 2020)– we now include a comparison to these 23 genes and reference the study. We have also added more direct reference to Carrasco et al. to ensure that this paper is properly recognized. We would like to emphasize, however, the difference between a gene showing regulated AS and APA (as identified from short read data) and the demonstration of connectivity between exon choices and alternative 3'UTRs that can only be determined using long reads. We use PL-Seq to show that *Dscam1* is not the only gene to show coordination of AS and APA. We find 22 others– we do not claim that the coordination is particularly widespread, but we do offer reasons why the 23 events we identify are likely an underestimate of the global impact of coordinated AS and APA. See also comment R2-3.

R1-3:

Figure 3 results: More statistics on the efficiency of PL-seq in pulling down full-length isoforms should be provided. For example: transcript coverage metrics as addressed in Sonesson et al., 2019; read length before and after filtering; junction coverage; % of splice signals and exon-intron junctions.

We thank reviewer for this suggestion. We have included the reads origin analysis in Fig. 3a to show the efficiency of PL-Seq method. Also, the gene body coverage, the read length distribution, the splicing events and junctions from libraries with/without pulldown enrichment are now included in Fig. 3c, d and supplementary Fig. 4. Note that since PL-Seq is probe-base method, these profiles are highly dependent on the target genes analyzed.

R1-4:

Figure 3: The approach the authors use to show exon inclusion within 3'UTR isoforms in the parsed long reads, could lead to confusion. For example, the total rate (in absolute terms) of exon production could still be higher for the short 3'UTR isoform. Authors show tracks of different scales; this enhances the view for specific exons that may not be necessarily be coupled to the 3'UTR, and should be avoided.

Thank you for this comment. We 100% agree with the reviewer that our data shows exon inclusion per 3'UTR isoform but does not quantify absolute exon production levels. Inherent to this technique, or any other that employs long reads, is that events which span more nucleotides (i.e. a cassette exon to a much longer 3'UTR) will be less represented than an equivalently expressed event that spans less nucleotides (i.e. cassette exon to a short 3'UTR). We present the data scaled because there is usually many fold more reads for a short 3'UTR isoform compared to a long 3'UTR isoform. If we compared on the same scale, the alternative splicing event for the long 3'UTR isoform would be impossible to see for many of the genes. We have now added the individual read tracks from different 3'UTR isoforms of *Dscam1* to help readers to appreciate the splicing CE pattern in each isoform (Fig. 2c). We are happy to add more of the individual read tracks if requested, in supplementary data.

R1-5:

For an accurate representation of the exon-3'UTR link, it would be helpful to compare the total abundance of a given exon to the overall gene expression.

Thanks for this comment. We have added the PSI from all transcripts (including all different 3'UTR isoforms) in Fig. 6c in the revised manuscript. If there are additional specific plots the reviewer suggests we are happy to provide them.

R1-6:

*Figure 6 results: from the absence of an exon from the short, but not the long 3'UTR isoform in *Dscam1* mRNAs, the authors conclude that "3'UTR isoform content or choice impacts Elav-mediated alternative splicing of exon 23". I disagree with this interpretation. The results show that the two events are linked, but not how. The causation could be the other way around, i.e., alternative splicing leads to APA, or, perhaps the most plausible explanation, ELAV binding causes both AS and APA on the same transcript.*

We 100% agree with this comment and have modified statements that make this overinterpretation in the paper. We do provide new evidence for specific cases (*Dscam1* and *Khc-73*) that 3'UTR choice impacts the upstream splicing from our new CRISPR deletion experiments (Fig. 5). However, we are more careful in our interpretation of these results. See also comment **R3-2**.

R1-7:

A de novo assembly of ELAV-dependent isoforms found only in the mutant would add more interesting insights of isoform regulation and ELAV-dependent 3'UTR choice.

We thank the reviewers for this suggestion and appreciate their interest in it. As PL-Seq uses a probe-base pulldown strategy to enrich transcripts from the genes of our interest, the PL-Seq data we collected in *elav* mutant samples are from only 27 genes. We examined the data of these genes and did not find high-confidence new isoforms that only appear in *elav* mutants. For us to pick up new isoforms with confidence, we might need to switch to the other long-read sequencing platform PacBio, which provides reads with much higher accuracy.

R1-8:

The elav5 molecular study lacks quality controls that support ELAV and control datasets are of good quality and comparable.

Thank you for this comment. PL-Seq is probe-base technique and pipeline, so the other quality control work for short-read RNASeq data cannot be applied to PL-Seq data. We use 3'UTR UTR isoform specific PSI for PL-Seq data analysis, which means exon inclusion events are already normalized to all the reads spanning this exon. For PL-Seq experiments performed on *elav* mutants and control embryos, we included probes for *elav* and *fne* to monitor the quality of our samples. All those from the *elav* mutant showed near zero expression of *elav* and high expression level of the *fne* mini-exon which is activated upon the loss of *elav* (Carrasco et al., 2020).

R1-9:

More insights into the regulation would have been desirable, for example by complementing this analysis with molecular studies (for example, ELAV CLIP and iCLIP studies are available) or functional studies (deletion of one or more CE to assess effect on associated 3'UTR).

Thank you for these suggestions. As suggested, for functional studies, we generated new CRISPR mutants skipping CE exclusively in *Khc-73* and *Dscam1* genes (Fig 5). Both mutants did not show any evidence for CE-splicing-affected 3'UTR choice (Fig. 5). In contrast, we found that deletion of long 3'UTRs for these same genes disrupted CE splicing of upstream exons (Fig. 5). Also see comment **R1-1 & R1-6**.

We have added more detailed analysis based on PL-Seq data collected from *elav* mutant and control embryos in Fig. 6c. We now demonstrate clearly how PL-Seq can distinguish CE splicing patterns for specific 3'UTR isoforms (CE PSI_{LONG}, CE PSI_{SHORT}) and how this differs from total CE splicing (CE PSI_{ALL}). The implication for this is that changes in CE splicing after an RBP knockdown (or other manipulations) might be masked when one only detects the equivalent of CE PSI_{ALL} from short read RNA-Seq or RT-PCR splicing assays. We present the examples of *Khc-73*, *Dys*, and *Calx* which have no change in CE PSI_{ALL} upon *Elav* loss, but do exhibit significant changes in CE PSI_{LONG} and/or CE PSI_{SHORT}.

As suggested, we examined existing ELAV embryo CLIP data for insights into regulation. We observed ELAV peak enrichment near proximal polyA sites as has been previously reported for ELAV regulated APA genes (Carrasco et al., 2020). However, when it came to the Elav regulated cassette exons of interest, there was a mixed bag of results. For *Dscam1* and *Khc-73* we observed peak enrichment for the 3'UTR connected CEs in the upstream intron, but not for other Elav regulated splicing events (e.g. *stai*, *Dys*, *Eip63E*, *Calx*, *par-1*) (data not shown).

Another CLIP dataset exists for overexpressed ELAV in S2 cells (Lee et al., Plos Genet, 2021). This paper focused on alternative splicing and did find that genes regulated by ELAV exhibited an enrichment for ELAV CLIP tags. They also reported a mild enrichment of ELAV CLIP peaks in intronic regions surrounding alternative exons that are skipped in the presence of ELAV. Again, a clear picture does not arise from the Elav binding data when it comes to our genes of interest in this work. The pattern of intronic enrichment around skipped exons holds for *Dscam1* in the embryo ELAV CLIP data, but we find the opposite for *Khc-73*. We do not feel that adding our re-analysis will build upon what has already been reported.

Minor comments:

R1-10:

Figure 1 results: To identify neuronal alternative CE AS, much better datasets than those used by the authors are now available: in sorted neurons and other cell populations (McCorkindale et al., 2019), and in elav mutant neuronal tissues (Wei et al., 2021). Figure 1 is overall redundant with previous studies.

Thank you for these suggestions. We used this old dataset because it has more replicates for each condition than the newer datasets that the reviewer pointed out, which is important for our analysis. We use t-test to call APA regulation from dPAU values. The more replicates it includes, the more power it had to call out a significant event. In Fig. 1 we need the analysis to obtain our list of AS-APA genes for determining a subset of genes to target with the PL-Seq method. This analysis shows that the 3'UTR lengthening genes are significantly associated with co-regulated CE events in the same genes, which is a very importance basis for this work and has not been uncovered before. We find significant coordination of AS and APA from short read data of not only early vs late embryos (Fig. 1E), but also head vs ovary (Fig. S2), and L1CNS Control vs *elav,fne* (Fig. 6A). Previous studies examining mouse ES to neuron differentiation did not find a significant association of AS with APA using the same statistical approach (Ha et al., Genome Biol., 2018). See also comment **R2-3**.

We use fold change of distal polyA site usage (dPAU) specifically to obtain the list because it is very sensitive to the change of relatively lowly expressed long 3'UTRs, which is the case for many of the genes in embryos. We do not emphasize the analysis in the revised manuscript, because we agree with the reviewer there is not much novelty in it compared to the lists of genes described in Carrasco et al., 2020 and Wei et

al., 2020. But the dataset meets our needs here, and provides the interested reader with an approach to use existing short read RNA-Seq data for obtaining candidate AS-APA regulated genes.

R1-11:

The abbreviation CE for cassette exon is not explicitly stated

Thank you, this is now addressed.

R1-12:

Typos are found throughout the text ("publically" at beginning of results section; and others).

Thank you, this is now addressed.

R1-13:

The way the results are presented could be improved to increase readability. For example, in Fig. 4c, numbers/proportions/ratios should be provided instead of tables with yes/no.

Thanks for the comment. We now include all the PSI values from late-stage embryos and adult heads in supplementary Fig. S5.

Reviewer #2 (Remarks to the Author):

R2-1:

In their manuscript Zhang and co-workers essentially report a novel method ('PL-Seq') to elucidate the link between alternative splicing and 3'UTR isoforms. Using this probe-based cDNA pulldown approach coupled to Nanopore sequencing, the authors identify more than 20 genes that exhibit alternative splicing linked to 3'UTR isoforms in neuron-enriched Drosophila tissues. They also show that the neuron-specific RNA-binding protein ELAV plays a role in this coordinated AS-APA regulation in Drosophila.

The variant of direct long-read sequencing introduced here is interesting and versatile. It helps to illuminate the landscape of co-regulated RNA processing and to decipher underlying mechanisms. The paper also provides a script on github for the PL-Seq workflow.

While I see a clear methodological advance, the underlying mechanisms and resulting biology of the coupling between AS and APA, which is indeed striking, remain somewhat vague.

We were pleased to read that the reviewer found the method interesting and versatile. We agree that the underlying mechanisms remain vague, but we make small steps toward understanding it in this revised manuscript.

Specific points:

R2-2:

What is the function of the transcripts co-regulated by AS and APA? Where do such RNAs localize in neuronal cells? Any obvious biological function that can be aligned to this type of regulation inferred from sequence, encoded protein domains etc? Does the 'dual specification' regulate spatially restricted protein functions in neuronal cells?

Thank you for the comment. We have added Gene Ontology analysis in Fig. 1f. It reveals multiple categories of enrichment for the 58 genes, including an enrichment for the molecular functions of phosphatase activity and receptor binding and enrichment in the biological processes of axon guidance. The long 3'UTR is usually only highly expressed in neuronal cells but the shorter isoforms could be highly expressed in other types of cells. We did not find commonalities in the protein domains included or excluded from the cassette exons that show bias for long or short 3'UTR by PL-Seq. This is from a small set of genes however, so we omit making a statement on it in the manuscript.

R2-3:

Overall, the number of coupled AS-APA events seems surprisingly small (given the enormous diversification of transcripts in Drosophila such as DSCAM etc). Is this limited extent specific to Drosophila? How does this compare to vertebrates?

Thank you for the comment. We have set very stringent cutoffs for the analysis of coupled AS-APA events to obtain the genes with high confidence. For example, to get 3'UTR lengthening genes, FoldChange (dPAU) >2 and adjusted p value <0.05 were set as the cutoff. Only long read sequencing performed transcriptome-wide at unprecedented depths could begin to really assess the scope of coordinated AS and APA. We admit, our paper is perhaps awkwardly positioned – our long read sequencing is not a transcriptome wide study of these connected events, and it is also not a study of just one or a few genes.

Previous studies examining mouse ES to neuron differentiation did not find a significant association of AS with APA (Ha...Morris, Genome Biol., 2018). We repeated this analysis and came to the same conclusion (Figure R1C below, not in manuscript). However, there were 115 candidate genes that exhibit regulation of both AS and APA during neuronal differentiation. So, using the same parameters for AS and APA analysis, a widespread scope for potential connected AS and APA was not identified.

Still, this does not mean that deeper data or data from mouse embryonic or neural development in vivo would not reveal an enrichment. See also comment **R1-10**.

Using PL-Seq, we were able to confirm connectivity of a CE to alternative short, medium, and long 3'UTRs for the *Endov* gene (Figure R1 D-E below, not in manuscript). We have not found a good way to integrate this data with the current manuscript, but present it here for the reviewer's interest.

Figure R1 (For reviewer's information– not in manuscript). 3'UTR linked CE splicing in mouse ES cell derived neurons. (a) Short read RNA-Seq analysis of mouse ES cell derived neurons (DIV7) versus undifferentiated ES cells (DIV-8) shows that 1537 genes exhibit significant 3'UTR lengthening. **(b)** Distribution of PSI change (PSI (neurons) – PSI (ES cells)). Horizontal dash lines in (a) indicate adjusted $p=0.05$, and in (b) indicate adjusted FDR=0.05. Vertical dash lines in (a) indicate FC=0.5 (left) and FC=2 (right), and in (b) indicate $\Delta\text{PSI}=-0.2$ (left) and $\Delta\text{PSI}=0.2$ (right). **(c)** Fisher's exact test shows that in mouse neurons vs ES samples, 3'UTR lengthening genes are not significantly associated with regulated CE events. **(d)** *Endov* exon 4 splicing pattern of short, medium and long 3'UTR reads in mouse ES cell derived neurons (DIV7) as shown by PL-Seq coverage tracks. **(E)** PL-Seq quantification of *Endov* exon 4 PSI shows that CE PSI is significantly higher in short 3'UTR isoforms when compared to medium and long 3'UTR isoforms in mouse ES cell derived neurons (DIV7). Paired two-tailed t-test. Data is shown as Mean + SEM. * indicates $p < 0.05$. $n=4$.

R2-4:

The abstract seems ambivalent: Is the focus on the method or the biology?

Thanks for this comment- we agree and have modified the abstract along with the text. Our focus now is on the utility of the method to quantify alternative splicing of CEs within long vs short 3'UTR isoforms.

R2-5:

Minor:

Introduction 1st paragraph, result section 1st paragraph: Recently, PCF11 was shown to be a quintessential regulator in neuronal development by regulating 3'UTR isoforms (PMID: 30552333). Is there evidence that PCF11 plays a role in the model used here?

Thank you for pointing out this. We have modified the introduction to cite this work. In *Drosophila*, Elav/Fne expression can explain almost all neural 3'UTR lengthening events while PCF11 regulates about 1/4 of APA events (Ogorodnikov et al., Nat. Commun., 2018), so in our opinion it is unlikely that PCF11 (conserved between *Drosophila* and mammals) is playing an analogous role in mammals. Hu proteins are candidate factors for accounting for neural-specific 3'UTR lengthening in mammals, and we have cited this in the introduction as well.

R2-6:

Introduction 1st paragraph: The translational significance of 3'UTR regulation could be strengthened (i.e. disease relevance)

Thank you for the comment. We have added statements on disease relevance for 3'UTR APA in the introduction and appropriate references.

R2-7:

Introduction 3rd paragraph, result section 1st paragraph: Acronym 'CE' page 2 and 3 not explained

Thank you, this is now addressed.

Supplementary Figure 1: legend title missing

Thank you, this is now addressed.

Abstract 2nd sentence: remainder of a previous reference

Thank you, this is now addressed.

Reviewer #3 (Remarks to the Author):

R3-1:

This timely manuscript investigates linked RNA processing events, namely coordinated alternative splicing (AS) with alternative polyA site usage (APA). The study follows up a

previous report by the same authors (Zhang, Z. et al. 2019. *Elav-Mediated Exon Skipping and Alternative Polyadenylation of the Dscam1 Gene Are Required for Axon Outgrowth*. *Cell Rep* 27, 3808-3817.e3807, doi:10.1016/j.celrep.2019.05.083) showing that changes in *Dscam* alternative splicing and APA as development proceeds is coordinated by the RNA binding protein ELAV, which is also investigated here. A series of papers from the Hilgers lab on these topics previously linked APA regulation by ELAV to transcriptional pausing, also in the fly nervous system. The major advance comprised in this paper is a targeted survey of AS and APA events in 31 genes using long read sequencing, a third generation RNA-seq method that enables detection of AS and APA status in the same long read. Despite the rigor of some of the data and their analyses, the present manuscript lacks biological significance and mechanistic insight. For example, the developmental setting is not exploited except as a source for library preparation. Given the previously published body work, the major gap in knowledge would seem to be the generality and/or mechanistic features that coordinate AS and APA choices. The manuscript does not do this. We make the following suggestions:

Thank you for these comments. We were pleased that the reviewer recognizes the utility of the method to detect AS and APA in the same read and the timeliness of the study. We hope that the data added in Figures 5 and 6 is viewed as a start to address the major gaps in mechanism.

Major points

R3-2:

1. *The findings lack novelty and mechanism*
The conceptual framework ignores obvious mechanistic features that could be tested. The authors completely avoid the question of whether APA site choice determines AS, or if it is the other way around. Most readers will know that 3' end cleavage of mRNA at the polyA site is, by definition, a cotranscriptional event that initiates transcription termination. They will also know that most introns are spliced co-transcriptionally in many species, including fly (papers by Rosbash and Carmo-Fonseca). Analysis of nascent RNA, rather than polyA+ mRNA, would answer the question of whether AS occurs before 3' end cleavage. If the 3'UTR affects upstream splicing, then these splicing events must be post-transcriptional; this would be an interesting and novel finding that would reflect a mechanistic feature. Previous studies also indicate coordination between splicing and 3' end cleavage, which are in some cases mediated by specific splicing regulators. Finally, generality of the reported coordination between AS and APA also seems to be lacking. Figures 1E and 6C indicate that many APA events are not coordinated with AS and vice versa. In this regard, figure 6F is not very helpful.

We thank the reviewer for the comments and suggestions. We are pleased to report that we have been able to directly address “the question of whether APA site choice determines AS, or if it is the other way around”. We generated CRISPR mutants for

Dscam1 and *Khc-73*, deleting the CEs and their flanking introns. We found, by qRT-PCR which is very sensitive for detecting 3'UTR length, that there was no change of distal polyA site usage in these mutants (Fig. 5c, d). In contrast, when we delete long 3'UTRs for *Dscam1* and *Khc-73*, we find CE splicing in the remaining short 3'UTR isoform is significantly altered, according to PL-Seq. See also comment **R1-1 and R1-6**.

In response to the important comments on the nature of the CE splicing being co- or post-transcriptional, we tried to investigate at the level of pre-mRNA by adjusting our PL-Seq method to start with nascent RNAs for 3' end primed reverse transcription. This would address whether polyA tail formation on the long 3'UTR can begin before splicing of upstream CEs is complete. Unfortunately, we did not obtain data of sufficient quality to make firm conclusions— we had difficulties to determine if a read is truncated or not at the 3' end when it lacks a poly A tail (i.e., whether it falls from the pore during the sequencing or it is internally primed). We may need to switch to PacBio platform to generate reads with higher accuracy and change our pipeline for downstream analysis.

Nascent RNAs including 3'UTR sequence are much longer than mature mRNAs, these are inherently much more difficult to quantify by long read sequencing. We have also had difficulties associated with obtaining high quality nuclei from embryonic neurons to obtain enough intact nascent RNA transcripts. We very much agree with the reviewer that determining the timing of the 3' end cleavage of the long 3'UTR relative to the cassette exon regulation is critical for understanding the mechanism. We have removed statements that are overly speculative when it comes to whether 3'UTR content or processing is determining CE regulation. Also see comment **R1-6**.

R3-3:

2. Methods and Data analysis.

Because the authors did not achieve a depth of coverage that allowed detailed analysis of the desired genes, they adapted the library prep to include enrichment of the cDNAs of interest using biotinylated probes; following this step, the captured cDNAs are reamplified and subjected to final library preparation and MinION sequencing. They dub this method Pull-a-Long-Seq (PL-seq). PL-seq is easier to multiplex than PCR, so we would consider it an advantageous variation on targeted long read sequencing. Given this, the methods and data analyses must be crystal clear. They state, “Two to five biotinylated probes were designed to target constitutive exons and universal 3'UTR sequences for each targeted gene”. This is a far lower density than traditional Capture-seq, but what is the density of probes? It is also unclear if they are only targeting one strand or both of each cDNA. What is the efficiency of targeting in terms of yield? Does this procedure introduced in the middle of the protocol introducing biases? It seems like it would be possible to address this by comparing long reads from a highly expressed gene that are not enriched versus enriched.

We thank the reviewer for the comments and suggestions. We now include more introduction and discussion to compare our PL-Seq method with other similar long-read sequencing methods. It is not only easier to multiplex than PCR, it also provides isoform

level quantification of CE splicing, as we show in Fig. 6c. For probe density, we include the probe sequences for every experiment in supplementary table 5. The probes are single stranded 5' biotinylated oligonucleotides, and they are designed to target the cDNA without regard to plus or minus strand. We have clarified these details in the methods section. Although the density is much lower than traditional Capture-seq, it works very well without introducing biases as we show for the example of *stai* (Fig. 3b). The efficiency of targeting is shown in Fig. 3a, and we can achieve several hundred times of enrichment in an experiment performed with probes targeting 15 to 30 genes.

R3-4:

We would like to see some of the actual reads and not just a pile up of reads. We find it odd that the data shown (e.g. Fig 2C) are so block-like. Long read sequencing usually includes a lot of incomplete reads. Have these been filtered out? Also, the data processing section is inadequate. What are the minimap parameters? Did they filter out PCR duplicates?

We thank the reviewer for this comment. The data is block-like because the coverage tracks we show are only of reads that have passed and filtering and have been assigned to either short or long 3'UTR groups. Incomplete reads that cannot cover upstream/downstream constitutive exons and the ends of 3'UTRs were filtered out as shown in Fig. 2b. We have updated the data processing methods section. Minimap2 parameters we utilized in this work is included in the methods now. We now show individual reads for *Dscam1* in embryos in Figure 2c.

R3-5:

Figure legends do not describe the data panels and need to be improved. The text currently provided is very interpretive, telling us what the data show rather than what the data are. Some of the figures themselves could be improved. Figure 2A is too simplistic and could be removed; the use of the word "linked" is a stretch because these are not necessarily functional links but correlations. Figure 3C, please indicate that this is a zoom in of the gene shown in B. Otherwise, it is confusing that the shape of the read density is different.

Thank you for the comments and suggestions. We have rewritten the figure legends part to make sure they are concise and not over-interpretive. We also removed the old Fig. 2a per this comment. We use the word "connected" to describe the 3'UTR dependent CE splicing pattern difference as it is intramolecularly connected as revealed by our long-read data. The figure in question is now modified to clarify that it is a Zoomed in view as requested (Fig. 2d).

R3-6:

On the bottom of the introduction page, it is stated: "Here we find that 80% of genes examined exhibit 3'UTR linked AS in neuron-enriched tissues". This compelling metric is

never mentioned again. Where does this come from? Is it globally analyzed genes? Or is this from the targeted 31 genes? If targeted, then the 80% statement is a bit misleading; 31 cherry-picked genes do not invite generalization.

Thank you for the comment. We agree that our PL-Seq analysis of 31 genes from our AS-APA gene list does not invite generalization. 80% previously referred to the 23 out of 28 AS-APA genes that we performed PL-Seq analysis and obtained enough reads to perform downstream analysis. We have thus changed the statement to “identify 23 genes” instead of 80%. See also comment **R1-2**.

R3-7:

Minor points:

Bottom third of the first page of the results: Fig 1C is about APA. These data are also in D but the numbers do not match up. Is the citation of the figure incorrect and there is another figure somewhere?

Thanks for this comment. Fig. 1c is about APA, but Fig. 1d is about the CE splicing events in the 232 genes that have been detected as 3'UTR lengthening genes during embryonic development in Fig. 1c. We have removed the numbers in Fig. 1d to avoid confusion.

R3-8:

Typo: “upstream constitute exon”

Thank you, this is now corrected.

R3-9:

Fig 1B: there is a green box around one line of data; what is that?

The green box was used to highlight the embryonic timepoint we used for our experiments. It has now been removed to avoid confusion.

R3-10:

Figure 4C contains the same data and A+B. We suggest moving C to the supplement, since significance is only indicated with “Yes” or “no”. The data in A&B are more quantitative. Some of the examples from Fig 5 could be put into Fig 4. It is not necessary to have the extra figure.

Thanks for the comment. We have added all the values requested and moved it to supplementary Fig. 5. See also comment **R1-13**.

R3-11:

Figures are not correctly referenced in the text (i.e. in-text references do not point to the correct data).

Thank you, this is now corrected.

REVIEWER COMMENTS

Reviewer #1 (Remarks to the Author):

This article investigates the potential coordination between alternative splicing (AS) and alternative polyadenylation (APA) in the *Drosophila* nervous system. Using a combination of cDNA pull-downs and long-read sequencing, the authors develop a new experimental/computational method called PL-seq. The authors show evidence of coupling between exons and 3'UTRs in genes that undergo lengthening during development in an ELAV-dependent manner. The revised manuscript includes new data and analyses, including CRISPR loss-of-function alleles of specific exons found to be associated with 3' ends, and 3'UTR mutants. We still think that the study provides interesting results. However, compared to the previous version of the paper, the revisions have not sufficiently increased conceptual or technological novelty, or mechanistic insight, which were our major concerns.

Actually, the new data raised additional concerns. Most importantly, we find the main claim (quantitative assessment of connectivity between AS and APA by PL-seq, and inference of causality between the two) insufficiently supported by the data.

1) One major concern, most prominently visible in Fig. 3b and 3c, is that the method shows an important 3'-end bias, which biases detection towards splicing that occurs adjacent to the 3'UTR. Given this bias and different lengths of 3'UTRs, conclusions about differential exon inclusions depending on 3'UTR length may be problematic. 2) The deletion of an alternative exon caused no change in APA; but conversely, a change in APA (3'UTR deletion) caused a change in the PSI, which the authors interpret as a causative effect of APA on splicing. However, there are other more obvious explanations for the effects seen upon 3'UTR depletion. For example, post-transcriptional regulation (e.g., 3'UTR loss causes loss of stability of the RNA isoform containing the exon in question).

Reviewer #2 (Remarks to the Author):

I appreciate the authors' effort to provide further mechanistic detail and biological significance. This greatly improved the current manuscript and illustrates that PL-Seq is a versatile tool to further explore the relationship and mechanisms underlying the functional link between alternative splicing and polyadenylation.

I noticed that the experiments on the causal relationship between splicing and alternative polyadenylation are based on 3 and 4 experiments (Fig. 4c and d), with Fig. 4c showing a non-significant trend. Since there is some variability, I would recommend including more replicates to consolidate the conclusions drawn (on the absence of a splicing effect).

Further points:

- I would recommend including Figure R1 in this manuscript (and be it even as a supplemental Information).
- Lines 40-41: "Disruption of APA regulators is linked to human disease, in particular members of the Cleavage Factor I (CFI) complex 22-24".

Importantly this has not only been "linked to human disease", alterations in APA factors can even become non-genomic disease drivers (PMID: 30552333).

- Lines 47-48: "Many RBPs that were known to regulate AS have been found to be involved in regulation of APA30,31"

The referencing seems a bit biased. A recent systematic profiling employing loss of function studies indicated the effect of splicing on APA (PMID: 32976578). This needs to be included.

- Line 210: I assume this sentence refers to Fig. 4a (instead of Fig. 5a)

- Lines 319-320: "For example, alternative promoter usage has been previously connected to alternative polyA site usage⁵⁴, and enhancer activity/transcriptional activity has been shown to regulate APA^{55,56}."

See also comment above: A transcriptome-wide APA profiling in more than 170 experimental conditions indicates numerous processes to be involved, including epigenetics and transcriptional activities, starting from transcription initiation, elongation and termination (PMID: 32976578). This needs to be included.

Comments of Reviewer #2 on Reviewer #1 concerns:

Concern #1: Almost all techniques based on oligo d(T) priming and reverse transcription show a 3' end bias. The pulldown introduces another bias (Fig. 3 c,d). However, this is not entirely unexpected. The main question here is whether this confounder really matters.

Normally we look at comparative analyses of two (case vs control) or more conditions. In these cases, a bias is normalized. This is, for example, demonstrated in Fig 4 a-d, where the 16-18 hr embryos (left) are compared to the adult heads (right). The differences are obvious, and thus I do not agree with this point of critique.

Even more: When we look at the data in a non-comparative fashion as shown in Fig. 3e, we clearly see that there is significant variation of genes with CE PSI in the long 3'UTR and CE PSI in short 3'UTR, documenting a reciprocal relationship between CE and UTR length (left to right). This can also be seen in Fig. 3f – even when looking at data in a non-comparative fashion.

However, as said above, the experimental reality would be that we study the coupling between CE and APA in case and control and thus I do not at all see a problem with this bias (inherent to most techniques).

Concern #2: I am not sure whether I want to agree here with reviewer 3. Yes in principle, there might be alternative mechanisms (RNA turnover etc). However, the scenario, which reviewer 3 considers here to represent "more obvious explanations", is in fact super-complex (and I would say even more speculative). This would then require that the different UTR and the differentially included exon would kick on (of off) RNA turnover-mechanisms either in dual specification or in a mutually exclusive (compensatory?) manner. I would not rule out that such scenarios do not exist, but I do not want to speculate here further. The authors provide some interesting functional insights, and PL-Seq is now available for studying the underlying connections.

However I realized that the graph shown in Fig. 5c relies only on 3 data points, which the authors might want to improve to substantiate their conclusion on the splicing effect.

In conclusion, in my eyes this is a very valuable tool to further explore the relationship and mechanisms underlying the functional link between alternative splicing and polyadenylation. It's cost effective and a bioinformatics platform is included as well.

Reviewer #3 (Remarks to the Author):

We appreciate all of the changes made to the study in response to all of the reviewers' comments. In particular, Figure 5's experiments point to the biological relevance of the alternative events by showing that without this coordination, different isoforms are expressed. A major comment is that the conclusions drawn – that APA site selection feeds back to splicing – is not justified by the data. The mechanism MAY be a direct interaction between splice site and polyA site choice, but it could equally be do to transcript stability. In addition, we still do not know where in the cell the transcripts localize, if they are bound by miRNAs, how long the polyA tail is, etc. This is fine, but these alternatives must be mentioned in the last paragraph of the discussion and given equal weight. The reason is that mature mRNA is being analyzed that that leaves a big black box around both co-transcriptional and post-transcriptional events that affect steady state levels. Thus, mechanistic details are still missing; we are looking at correlations.

We additionally want to point out the following issues:

1. Throughout the figures, gene diagrams should be labeled to indicate the direction of the gene (arrow at TSS) and should include a scale bar (since gene length is potentially important re methodological constraints).
2. In Figure 2b, the third step from the top is missing a word (i.e., "Reads containing exon downstream of").
3. Could the authors add a statement clarifying why the stronger 3' end bias in the pulldown library is not a concern? The way they show their long reads in Fig 2c is not very informative in terms of looking at splicing events. Also, in almost all cases there are far fewer reads from long isoforms compared to short. It seems like the authors should be able to dig into the data to figure out why the control and pulldown disagree in Figure 3C. It is unclear how this plot relates to Fig 2C. This figure legend does not sufficiently describe what is plotted (it says "gene body coverage" without explaining what that means exactly). Is the PL dataset biased towards spliced transcripts? That would account for why the reads are shorter. Why is "control" a huge block over the whole gene body? Is it contaminated with DNA? Is the control total RNA or polyA+?
4. If they targeted genes that have alternative isoforms with varying 5' ends for pulldown, this may help explain the bias toward the 3' end. This looks to be true for the stai gene shown as an example. It's good that reads must contain at least one exon upstream of the cassette exon to be included for analysis. Their filtering steps outlined in Figure 2 are appropriate.
5. When referencing Figure 3b in the text and the figure legend, the authors refer to stai exon 10; however, Figure 3b shows stai exon 6. Why is this inconsistent?
6. Please note a recent publication in Cell must be referenced and discussed: Sites of transcription initiation drive mRNA isoform selection from the Hilgers lab, which includes APA site selection and correlations with TSSs in Drosophila, using long read sequencing.
<https://pubmed.ncbi.nlm.nih.gov/37178687/>

REVIEWER COMMENTS

Reviewer #1 (Remarks to the Author):

R1-main. This article investigates the potential coordination between alternative splicing (AS) and alternative polyadenylation (APA) in the *Drosophila* nervous system. Using a combination of cDNA pull-downs and long-read sequencing, the authors develop a new experimental/computational method called PL-seq. The authors show evidence of coupling between exons and 3'UTRs in genes that undergo lengthening during development in an ELAV-dependent manner. The revised manuscript includes new data and analyses, including CRISPR loss-of-function alleles of specific exons found to be associated with 3' ends, and 3'UTR mutants. We still think that the study provides interesting results. However, compared to the previous version of the paper, the revisions have not sufficiently increased conceptual or technological novelty, or mechanistic insight, which were our major concerns.

Actually, the new data raised additional concerns. Most importantly, we find the main claim (quantitative assessment of connectivity between AS and APA by PL-seq, and inference of causality between the two) insufficiently supported by the data.

>Thank you for pointing out these concerns. We trust that the main concerns of the quantitative assessment of connectivity between AS and APA by PL-Seq have now been resolved with the updated Figure 3c. This shows that there is no bias for the 3' end with PL-Seq over what occurs for the standard nanopore library preparation without pulldown when the same genes are plotted. See detailed response below (R3-3).

We agree that our mechanistic insight is limited with regard to how loss of long 3'UTR alters upstream alternative splicing for *Khc-73* and *Dscam1*. We are careful to not present a biased view on how this might be occurring (See Lines 380-394). Also see R3-main response.

R1-1) One major concern, most prominently visible in Fig. 3b and 3c, is that the method shows an important 3'-end bias, which biases detection towards splicing that occurs adjacent to the 3'UTR. Given this bias and different lengths of 3'UTRs, conclusions about differential exon inclusions depending on 3'UTR length may be problematic. 2) The deletion of an alternative exon caused no change in APA; but conversely, a change in APA (3'UTR deletion) caused a change in the PSI, which the authors interpret as a causative effect of APA on splicing. However, there are other more obvious explanations for the effects seen upon 3'UTR depletion. For example, post-transcriptional regulation (e.g., 3'UTR loss causes loss of stability of the RNA isoform containing the exon in question).

>Regarding the 3' bias, this has now been clarified in Fig. 3c. See R3-3 response clarification and R2 response to concern #1.

Regarding the interpretation of the change in splicing resulting from long 3'UTR deletion. We have been careful in the revised version not to over interpret these results as a causative effect

of APA on alternative splicing, and present multiple alternative interpretations in the discussion. See response to R3-main.

Reviewer #2 (Remarks to the Author):

R2-1. I appreciate the authors' effort to provide further mechanistic detail and biological significance. This greatly improved the current manuscript and illustrates that PL-Seq is a versatile tool to further explore the relationship and mechanisms underlying the functional link between alternative splicing and polyadenylation.

>Thank you very much.

R2-2. I noticed that the experiments on the causal relationship between splicing and alternative polyadenylation are based on 3 and 4 experiments (Fig. 4c and d), with Fig. 4c showing a non-significant trend. Since there is some variability, I would recommend including more replicates to consolidate the conclusions drawn (on the absence of a splicing effect).

> Thank for the suggestion. We have repeated these RT-qPCR experiments with new biological replicates samples (n=6). The variability is reduced and conclusions remain the same (absence of a splicing effect on APA) (Fig. 5c,d).

Further points:

• **R2-3.** I would recommend including Figure R1 in this manuscript (and be it even as a supplemental Information).

> Thank for the suggestion. We have added this figure as Supplementary Figure 8 and integrate the finding into the results section (lines 308-319).

R2-4. Lines 40-41: "Disruption of APA regulators is linked to human disease, in particular members of the Cleavage Factor I (CFI) complex 22–24".

Importantly this has not only been "linked to human disease", alterations in APA factors can even become non-genomic disease drivers (PMID: 30552333).

> Thank you for the comment. The Ogorodnikov paper (PMID: 30552333) showed that expression levels of PCF11 were significantly lower in neuroblastomas of stage 4S (better prognosis) compared to stage 4 (poor prognosis) and that lower PCF11 was found to be associated with higher survival. Moreover, PCF11 knockdown altered global APA patterns and increased neuronal differentiation. Thus, PCF11 is involved in neuroblastoma progression. However, in this publication there is no claim made or direct evidence provided that PCF11 or other APA factors can serve as "non-genomic disease drivers". So, we prefer not to use that

term.

We have now referenced the Ogorodnikov paper in the context of a CFII component being involved in human disease (lines 44-46): “Disruption of APA regulators is involved in human disease, in particular members of the Cleavage Factor I and II complexes(14, 22-24)”

R2-4. Lines 47-48: “Many RBPs that were known to regulate AS have been found to be involved in regulation of APA30,31”

The referencing seems a bit biased. A recent systematic profiling employing loss of function studies indicated the effect of splicing on APA (PMID: 32976578). This needs to be included.

>Thank you for pointing this out. We have broadened the referenced papers with regard to this sentence, and include the suggested reference (Marini et al., 2021. PMID: 32976578) (lines 50-53).

R2-5. Line 210: I assume this sentence refers to Fig. 4a (instead of Fig. 5a)

>Yes, thank you, this has been corrected.

R2-6. Lines 319-320: “For example, alternative promoter usage has been previously connected to alternative polyA site usage⁵⁴, and enhancer activity/transcriptional activity has been shown to regulate APA^{55,56}.”

See also comment above: A transcriptome-wide APA profiling in more than 170 experimental conditions indicates numerous processes to be involved, including epigenetics and transcriptional activities, starting from transcription initiation, elongation and termination (PMID: 32976578). This needs to be included.

>Thank you for the comment. We have expanded the referencing and the sentence, and now include the Marini paper (PMID: 32976578). (Lines 346-349).

“Regulation of APA clearly involves more than the binding of RBPs and the core cleavage and polyadenylation machinery in the vicinity of polyA sites– roles for enhancer/transcription activity, DNA methylation, and specific chromatin remodeling proteins have emerged in recent years (28,58–61)”

Comments of Reviewer #2 on Reviewer #1 concerns:

Concern #1: Almost all techniques based on oligo d(T) priming and reverse transcription show a 3' end bias. The pulldown introduces another bias (Fig. 3 c,d). However, this is not entirely unexpected. The main question here is whether this confounder really matters.

Normally we look at comparative analyses of two (case vs control) or more conditions. In these cases, a bias is normalized. This is, for example, demonstrated in Fig 4 a-d, where the 16-18 hr

embryos (left) are compared to the adult heads (right). The differences are obvious, and thus I do not agree with this point of critique.

Even more: When we look at the data in a non-comparative fashion as shown in Fig. 3e, we clearly see that there is significant variation of genes with CE PSI in the long 3'UTR and CE PSI in short 3'UTR, documenting a reciprocal relationship between CE and UTR length (left to right). This can also be seen in Fig. 3f – even when looking at data in a non-comparative fashion.

However, as said above, the experimental reality would be that we study the coupling between CE and APA in case and control and thus I do not at all see a problem with this bias (inherent to most techniques).

Concern #2: I am not sure whether I want to agree here with reviewer 3. Yes in principle, there might be alternative mechanisms (RNA turnover etc). However, the scenario, which reviewer 3 considers here to represent “more obvious explanations”, is in fact super-complex (and I would say even more speculative). This would then require that the different UTR and the differentially included exon would kick on (of off) RNA turnover-mechanisms either in dual specification or in a mutually exclusive (compensatory?) manner. I would not rule out that such scenarios do not exist, but I do not want to speculate here further. The authors provide some interesting functional insights, and PL-Seq is now available for studying the underlying connections.

However I realized that the graph shown in Fig. 5c relies only on 3 data points, which the authors might want to improve to substantiate their conclusion on the splicing effect.

In conclusion, in my eyes this is a very valuable tool to further explore the relationship and mechanisms underlying the functional link between alternative splicing and polyadenylation. It's cost effective and a bioinformatics platform is included as well.

Reviewer #3 (Remarks to the Author):

R3-main. We appreciate all of the changes made to the study in response to all of the reviewers' comments. In particular, Figure 5's experiments point to the biological relevance of the alternative events by showing that without this coordination, different isoforms are expressed. A major comment is that the conclusions drawn – that APA site selection feeds back to splicing – is not justified by the data. The mechanism MAY be a direct interaction between splice site and polyA site choice, but it could equally be do to transcript stability. In addition, we still do not know where in the cell the transcripts localize, if they are bound by miRNAs, how long the polyA tail is, etc. This is fine, but these alternatives must be mentioned in the last paragraph of the discussion and given equal weight. The reason is that mature mRNA is being analyzed that that leaves a big black box around both co-transcriptional and post-transcriptional events that affect steady state levels. Thus, mechanistic details are still missing; we are looking at correlations.

>Thank you for the comments. We have avoided a biased interpretation in the results and discussion that might suggest that a direct interaction or mechanistically proved communication between splice sites and polyA site choice has been established. We have not uncovered such a detailed mechanism. We have tried to suggest alternative explanations in the last paragraph of the discussion (Lines 389-394). We have also explicitly stated that our study measures transcript isoforms at steady state levels, and future investigation that incorporates long read sequencing of nascent RNA with metabolic labeling might help shed light on mechanism.

We additionally want to point out the following issues:

R3-1. Throughout the figures, gene diagrams should be labeled to indicate the direction of the gene (arrow at TSS) and should include a scale bar (since gene length is potentially important re methodological constraints).

>Thank you for the suggestion. We have now included direction and scale bar in all figures. When a TSS is in the gene model, we have also labelled the TSS. Fig 2d, 3b, 4c, 4d.

R3-2. In Figure 2b, the third step from the top is missing a word (i.e., “Reads containing exon downstream of”).

>Thanks for noticing this. It has now been corrected.

R3-3. Could the authors add a statement clarifying why the stronger 3' end bias in the pulldown library is not a concern? The way they show their long reads in Fig 2c is not very informative in terms of looking at splicing events. Also, in almost all cases there are far fewer reads from long isoforms compared to short. It seems like the authors should be able to dig into the data to figure out why the control and pulldown disagree in Figure 3C. It is unclear how this plot relates to Fig 2C. This figure legend does not sufficiently describe what is plotted (it says “gene body coverage” without explaining what that means exactly). Is the PL dataset biased towards spliced transcripts? That would account for why the reads are shorter. Why is “control” a huge block over the whole gene body? Is it contaminated with DNA? Is the control total RNA or polyA+?

> Thank you for the comments. Regarding the display of long reads in Fig. 2c– this is showing the actual individual reads, which was requested specifically in the first review. If one focuses on exons 12 and 15 in the gene model, then scans down, it is evident that there is much less inclusion of exons 12 and 15 for the short 3'UTR reads. Fig. 2d provides the read pile up where the biased inclusion of exons 12 and 15 in the long 3'UTR isoform is more clear. We have not altered this figure any further.

Regarding Fig. 3c, we apologize for any confusion from the previous plot. Fig 3c now shows “no-pulldown”, “pulldown”, no-pulldown (15 target genes)”, “pulldown (15 target genes)”. There is

no 3' bias that is introduced by the pulldown for the 15 targeted genes compared to no-pulldown condition (compare dotted lines). There are far fewer reads from long compared to the short 3'UTR isoforms because of the inherent nature of long read sequencing. Very long reads will be of lower abundance compared to short reads. This is always the case across platforms, unless one size selects for particular transcripts or cDNAs prior to library preparation. The figure legend and description has been clarified.

R3-4. If they targeted genes that have alternative isoforms with varying 5' ends for pulldown, this may help explain the bias toward the 3' end. This looks to be true for the stai gene shown as an example. It's good that reads must contain at least one exon upstream of the cassette exon to be included for analysis. Their filtering steps outlined in Figure 2 are appropriate.

>Thank you. See earlier responses on 3' end bias. The genes targeted for pulldown were longer genes than found on average in the transcriptome. Also, standard nanopore sequencing performed transcriptome wide will cover transcripts from 5' to 3' end mostly just for shorter transcripts, and will miss longer transcripts such as those with long 3'UTRs. Stai generates short full length mRNAs (~1.8 to 3 kb) and is highly expressed, thus deep coverage from 5' to 3' end was possible.

R3-5. When referencing Figure 3b in the text and the figure legend, the authors refer to stai exon 10; however, Figure 3b shows stai exon 6. Why is this inconsistent?

>Thank you for noticing this. It has been corrected to exon 6.

R3-6. Please note a recent publication in Cell must be referenced and discussed: Sites of transcription initiation drive mRNA isoform selection from the Hilgers lab, which includes APA site selection and correlations with TSSs in Drosophila, using long read sequencing.
<https://pubmed.ncbi.nlm.nih.gov/37178687/>

>Yes, this important publication was published 2 months ago and we now discuss the findings extensively (Lines 74-75, Lines 342-346).